# MIXED PRECISION DNNS:
## ALL YOU NEED IS A GOOD PARAMETRIZATION

**Stefan Uhlich,**[*] **Lukas Mauch,**[*] **Fabien Cardinaux,**[*] **Kazuki Yoshiyama**
**Javier Alonso García, Stephen Tiedemann, Thomas Kemp**
Sony Europe B.V., Germany
`firstname.lastname@sony.com`

**Akira Nakamura**
Sony Corporate, Japan
`akira.b.nakamura@sony.com`

### ABSTRACT

Efficient deep neural network (DNN) inference on mobile or embedded devices typically involves quantization of the network parameters and activations. In particular, mixed precision networks achieve better performance than networks with homogeneous bitwidth for the same size constraint. Since choosing the optimal bitwidths is not straight forward, training methods, which can learn them, are desirable. Differentiable quantization with straight-through gradients allows to learn the quantizer's parameters using gradient methods. We show that a suited parametrization of the quantizer is the key to achieve a stable training and a good final performance. Specifically, we propose to parametrize the quantizer with the step size and dynamic range. The bitwidth can then be inferred from them. Other parametrizations, which explicitly use the bitwidth, consistently perform worse. We confirm our findings with experiments on CIFAR-10 and ImageNet and we obtain mixed precision DNNs with learned quantization parameters, achieving state-of-the-art performance.

## 1 INTRODUCTION

Quantized DNNs apply quantizers $Q : \mathbb{R} \to \{q_1, ..., q_I\}$ to discretize the weights and/or activations of a DNN (Han et al., 2015; Zhou et al., 2017; Li et al., 2016; Liu & Mattina, 2019; Cardinaux et al., 2018; Jain et al., 2019; Bai et al., 2018). They require considerably less memory and have a lower computational complexity, since discretized values $\{q_1, ..., q_I\}$ can be stored, multiplied and accumulated efficiently. This is particularly relevant for inference on mobile or embedded devices with limited computational power.

However, gradient based training of quantized DNNs is difficult, as the gradient of a quantization function vanishes almost everywhere, i.e., backpropagation through a quantized DNN almost always returns a zero gradient. Different solutions to this problem have been proposed in the literature:
A first possibility is to use DNNs with stochastic weights from a categorical distribution and to optimize the *evidence lower bound* (ELBO) to obtain an estimate of the posterior distribution of the weights. As proposed in (Jang et al., 2016; Maddison et al., 2016; Louizos et al., 2019), the categorical distribution can be relaxed to a concrete distribution – a smoothed approximation of the categorical distribution – such that the ELBO becomes differentiable under reparametrization.
A second possibility is to use the *straight through estimator* (STE) (Bengio et al., 2013). STE allows the gradients to be backpropagated through the quantizers and, thus, the network weights can be adapted with standard gradient descent (Hubara et al., 2016). Compared to STE based methods, stochastic methods suffer from large gradient variance, which makes training of large quantized DNNs difficult. Therefore, STE based methods are more popular in practice.

More recent research (Jain et al., 2019; Esser et al., 2019; Wang et al., 2018; Elthakeb et al., 2018) focuses on methods which can also learn the optimal quantization parameters, e.g., the stepsize, dynamic range and bitwidth, in parallel to the network weights. This is a promising approach as DNNs with learned quantization parameters almost always outperform DNNs with handcrafted ones.

---

[*]Equal contribution.

Recently, and in parallel to our work, (Jain et al., 2019) explored the use of STE to define the gradient with respect to the quantizers's dynamic range. The authors applied a per-tensor quantization and used the dynamic range as an additional trainable parameter also learned with gradient descent. Similarly, (Esser et al., 2019) learned the stepsize using gradient descent. However, neither of them learned the optimal bitwidth of the quantizers.

One approach was proposed in (Wang et al., 2018; Elthakeb et al., 2018). They learn the bitwidth with reinforcement learning, i.e., they learn an optimal bitwidth assignment policy. Their experiments show that a DNN with a learned and heterogeneous bitwidth assignment outperforms quantized DNNs with a homogeneous bitwidth assignment. However, such methods have a high computational complexity as the bitwidth policy must be learned, which involves training many quantized DNNs.

In this paper, we will use the STE approach and show that the quantizer's parameters, including the bitwidth, can be learned with gradient methods if a good parametrization is chosen. Specifically, we show that directly learning the bitwidth is not optimal. Instead, we propose to learn the stepsize and dynamic range. The bitwidth can then be inferred from them. Compared to (Wang et al., 2018; Elthakeb et al., 2018), our method has the advantage that training quantized DNNs has nearly the same computational complexity as standard float32 training.

The contributions of this paper are:

1. We show that there are three different parametrizations for *uniform* and *power-of-two* quantization and that, in both cases, one of them has gradients particularly well suited to train quantized DNNs. The other parametrizations have the problem of yielding gradients with an unbounded gradient norm and coupled components.

2. Using this parametrization, we are able to learn all quantization parameters for DNNs with per-tensor quantization and global memory constraints. We formulate the training as a constrained optimization problem, where the quantized DNN is constrained not to exceed a given overall memory budget, and show how to solve it in a penalty framework.

3. We confirm our findings with experiments on CIFAR-10 and ImageNet. For example, we train a heterogeneously quantized MobileNetV2 on ImageNet requiring a total of only $1.65$MB to store the weights and only $0.57$MB to store its largest feature map. This is equivalent to a homogenous $4$bit quantization of both weights and activations. However, our network learns to allocate the bitwidth heterogeneously in an optimal way. Our MobileNetV2 achieves an error of $30.26\%$ compared to $29.82\%$ for the floating point baseline. This is state-of-the-art for such a heavily quantized MobileNetV2.

We use the following notation throughout this paper: $x$, $\mathbf{x}$, $\mathbf{X}$ and $\boldsymbol{\mathcal{X}}$ denote a scalar, a (column) vector, a matrix and a tensor with three or four dimensions, respectively; $\lfloor . \rfloor$ and $\lceil . \rceil$ are the floor and ceiling operators. Finally, $\delta(.)$ denotes the Dirac delta function.

## 2 Choosing a quantization parametrization

Let $Q(x; \boldsymbol{\theta})$ be a quantizer with the parameters $\boldsymbol{\theta}$, which maps $x \in \mathbb{R}$ to discrete values $\{q_1, ..., q_I\}$. In this section, we compare different parametrizations of $Q(x; \boldsymbol{\theta})$ for *uniform quantization* and *power-of-two quantization* and analyze how well the corresponding straight-through gradient estimates $\partial_x Q(x; \boldsymbol{\theta})$ and $\nabla_\theta Q(x; \boldsymbol{\theta})$ are suited to optimize the quantizer parameters $\boldsymbol{\theta}$. Our key result is, that the training of quantized DNNs which learns both, the optimal quantized weights and the optimal quantization parameters $\boldsymbol{\theta}$, is very sensitive to the choice of the parametrization of the quantizers. From an optimization point of view, it is best to parametrize the quantizer $Q(x; \boldsymbol{\theta})$ with the stepsize $d$ and the dynamic range $q_{\max}$ as it leads to gradients with stable norms. Doing so, we can use standard gradient descent to learn the quantization parameters and do not need to use stochastic or reinforcement based algorithms, which are computationally expensive.

### 2.1 Parametrization and straight through gradient estimates

A symmetric uniform quantizer $Q_U(x; \boldsymbol{\theta})$ which maps a real value $x \in \mathbb{R}$ to one of $I = 2k + 1$ quantized values $q \in \{-kd, ..., 0, ..., kd\}$ computes

$$q = Q_U(x; \boldsymbol{\theta}) = \text{sign}(x) \begin{cases} d \left\lfloor \frac{|x|}{d} + \frac{1}{2} \right\rfloor & |x| \leq q_{\max} \\ q_{\max} & |x| > q_{\max} \end{cases}, \tag{1}$$

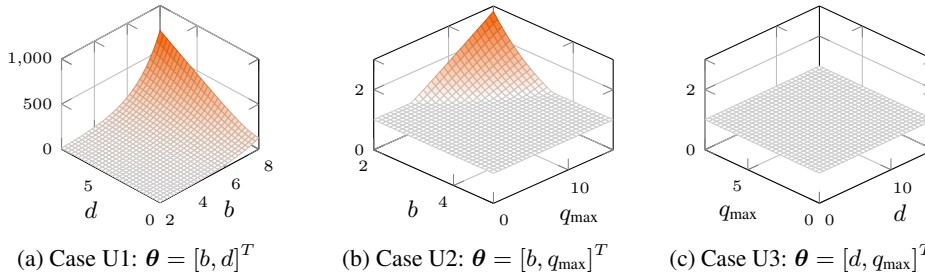

(a) Case U1: $\boldsymbol{\theta} = [b, d]^T$     (b) Case U2: $\boldsymbol{\theta} = [b, q_{\max}]^T$     (c) Case U3: $\boldsymbol{\theta} = [d, q_{\max}]^T$

Figure 1: Maximum gradient norm $\max_x \|\nabla_\theta Q_U(x; \boldsymbol{\theta})\|$. For "U1" and "U2" the maximum gradient norm can grow exponentially with varying bitwidth $b$ whereas it is bounded for "U3".

where the parameter vector $\boldsymbol{\theta} = [d, q_{\max}, b]^T$ consists of the step size $d \in \mathbb{R}$, the maximum value $q_{\max} \in \mathbb{R}$ and the number of bits $b \in \mathbb{N}$, $b \geq 2$ used to encode the quantized values $q$.

When training quantized DNNs, we want to optimize $Q_U(x; \boldsymbol{\theta})$ with respect to the input $x$ and the quantization parameters $\boldsymbol{\theta}$, meaning that we need the gradients $\nabla_x Q_U(x; \boldsymbol{\theta})$ and $\nabla_\theta Q(x; \boldsymbol{\theta})$. A common problem is, that the exact gradients are not useful for training. For example, $\partial_x Q_U(x; \boldsymbol{\theta}) = d \sum_{k=-2^{b-1}+1}^{2^{b-1}-2} \delta\left(x - d\left(k + \frac{1}{2}\right)\right)$ vanishes almost everywhere. A solution is to define the derivative using STE (Bengio et al., 2013), which ignores the floor operation in (1). This leads to

$$\partial_x Q_U(x) = \begin{cases} 1 & |x| \leq q_{\max} \\ 0 & |x| > q_{\max} \end{cases}, \tag{2}$$

which is non-zero in the interesting region $|x| \leq q_{\max}$ and which turned out to be very useful to train quantized DNNs in practice (Yin et al., 2019). In this work, we follow this idea and define the gradients $\nabla_x Q(x; \boldsymbol{\theta})$ and $\nabla_\theta Q(x; \boldsymbol{\theta})$, using STE whenever we need to differentiate a floor operation. We refer to this as *differentiable quantization* (DQ).

An important observation from (1) is that the parameters $\boldsymbol{\theta} = [d, q_{\max}, b]^T$ of a quantizer depend on each other, i.e., $q_{\max} = (2^{b-1} - 1)d$. This means, that we can choose from three equivalent parametrizations of $Q_U(x; \boldsymbol{\theta})$: Case "U1" with $\boldsymbol{\theta} = [b, d]^T$, case "U2" with $\boldsymbol{\theta} = [b, q_{\max}]^T$ and case "U3" with $\boldsymbol{\theta} = [d, q_{\max}]^T$. Interestingly, they differ in their gradients:

*Case U1:* Parametrization with respect to $\boldsymbol{\theta} = [b, d]^T$, using $q_{\max} = q_{\max}(b, d)$ gives

$$\nabla_\theta Q_U(x; \boldsymbol{\theta}) = \begin{bmatrix} \partial_b Q_U(x; \boldsymbol{\theta}) \\ \partial_d Q_U(x; \boldsymbol{\theta}) \end{bmatrix} = \begin{cases} \begin{bmatrix} 0 \\ \frac{1}{d} \end{bmatrix} (Q_U(x; \boldsymbol{\theta}) - x) & |x| \leq (2^{b-1} - 1)d \\ \begin{bmatrix} 2^{b-1} \log(2)d \\ 2^{b-1} - 1 \end{bmatrix} \operatorname{sign}(x) & |x| > (2^{b-1} - 1)d \end{cases} \tag{3a}$$

*Case U2:* Parametrization with respect to $\boldsymbol{\theta} = [b, q_{\max}]^T$, using $d = d(b, q_{\max})$ gives

$$\nabla_\theta Q_U(x; \boldsymbol{\theta}) = \begin{bmatrix} \partial_b Q_U(x; \boldsymbol{\theta}) \\ \partial_{q_{\max}} Q_U(x; \boldsymbol{\theta}) \end{bmatrix} = \begin{cases} \begin{bmatrix} -\frac{2^{b-1} \log 2}{2^{b-1}-1} \\ \frac{1}{q_{\max}} \end{bmatrix} (Q_U(x; \boldsymbol{\theta}) - x) & |x| \leq q_{\max} \\ \begin{bmatrix} 0 \\ \operatorname{sign}(x) \end{bmatrix} & |x| > q_{\max} \end{cases} \tag{3b}$$

*Case U3:* Parametrization with respect to $\boldsymbol{\theta} = [d, q_{\max}]^T$, using $b = b(d, q_{\max})$ gives

$$\nabla_\theta Q_U(x; \boldsymbol{\theta}) = \begin{bmatrix} \partial_d Q_U(x; \boldsymbol{\theta}) \\ \partial_{q_{\max}} Q_U(x; \boldsymbol{\theta}) \end{bmatrix} = \begin{cases} \begin{bmatrix} \frac{1}{d} \\ 0 \end{bmatrix} (Q_U(x; \boldsymbol{\theta}) - x) & |x| \leq q_{\max} \\ \begin{bmatrix} 0 \\ \operatorname{sign}(x) \end{bmatrix} & |x| > q_{\max} \end{cases} \tag{3c}$$

Fig. 1 shows the maximum gradient norm $\max_x \|\nabla_\theta Q_U(x; \boldsymbol{\theta})\|$ for the three parametrizations "U1" to "U3". For the parametrizations "U1" and "U2", $\max_x \|\nabla_\theta Q_U(x; \boldsymbol{\theta})\|$ can grow exponentially with varying bitwidth $b$ as $\partial_d Q_U(x; \boldsymbol{\theta}) \in [-2^{b-1} - 1, 2^{b-1} - 1]$ for "U1" and $\partial_b Q_U(x; \boldsymbol{\theta}) \in \left[-\frac{q_{max}}{2^{b-1}-1} \log 2, \frac{q_{max}}{2^{b-1}-1} \log 2\right]$ for "U2". This is not desirable when training quantized DNNs, because it will lead to large changes of the gradient norm and forces us to use small learning rates to

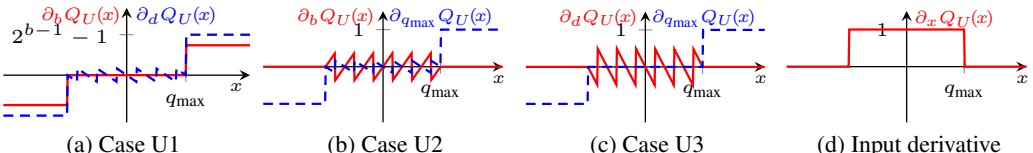

| (a) Case U1 | (b) Case U2 | (c) Case U3 | (d) Input derivative |

Figure 2: Partial derivatives of $Q_U(x; \boldsymbol{\theta})$ with respect to the input and the quantization parameters $d, q_{max}$ and $b$. Partial derivatives are coupled for "U1" and "U2" but are decoupled for "U3".

avoid divergence. However, parametrization "U3" does not suffer from such an unbounded gradient norm as both partial derivatives $\partial_d Q_U(x; \boldsymbol{\theta}) \in [-\frac{1}{2}, \frac{1}{2}]$ and $\partial_{q_{max}} Q_U(x; \boldsymbol{\theta}) \in \{-1, 1\}$ are bounded.

Fig. 2 shows the gradients for the parametrization "U1" to "U3". For parametrization "U3", the partial derivatives in $\boldsymbol{\nabla}_\theta Q_U(x; \boldsymbol{\theta})$ are *decoupled*, i.e., $\boldsymbol{\nabla}_\theta Q_U(x; \boldsymbol{\theta})$ is a unit vector, which either points only in the direction of $d$ if $|x| \le q_{max}$ or only in the direction of $q_{max}$, if $|x| > q_{max}$. We will show in Sec. 2.3 that this implies a diagonal Hessian, which results in a better convergence behavior of gradient descent. In contrast, both parametrizations "U1" and "U2" have partial derivatives that are coupled. In summary, this implies that parametrization "U3" is the best DQ parametrization.

Similar considerations can be made for the power-of-two quantization $Q_P(x; \boldsymbol{\theta})$, which maps a real-valued number $x \in \mathbb{R}$ to a quantized value $q \in \{\pm 2^k : k \in \mathbb{Z}\}$ by

$$q = Q_P(x; \boldsymbol{\theta}) = \text{sign}(x) \begin{cases} q_{min} & |x| \le q_{min} \\ 2^{\lfloor 0.5 + \log_2 |x| \rfloor} & q_{min} < |x| \le q_{max} \\ q_{max} & |x| > q_{max} \end{cases} , \tag{4}$$

where $q_{min}$ and $q_{max}$ are the minimum and maximum absolute values of the quantizer for a bitwidth of $b$ bit. Power-of-two quantization is an especially interesting scheme for DNN quantization, since a multiplication of quantized values can be implemented as an addition of the exponents. Using the STE for the floor operation, the derivative $\partial_x Q_P(x; \boldsymbol{\theta})$ is given by

$$\partial_x Q_P(x) = \begin{cases} 0 & |x| \le q_{min} \\ \frac{2^{\lfloor 0.5 + \log_2 |x| \rfloor}}{|x|} & q_{min} < |x| \le q_{max} \\ 0 & |x| > q_{max} \end{cases} . \tag{5}$$

The power-of-two quantization has the three parameters $[b, q_{min}, q_{max}] =: \boldsymbol{\theta}$, which depend on each other with the relationship $q_{max} = 2^{2^{b-1}-1} q_{min}$. Therefore, we have again three different parametrizations with $\boldsymbol{\theta} = [b, q_{min}]$, $\boldsymbol{\theta} = [b, q_{max}]$ or $\boldsymbol{\theta} = [q_{min}, q_{max}]$, respectively. Similarly to the uniform case, one parametrization ($\boldsymbol{\theta} = [q_{min}, q_{max}]$) leads to a gradient of a very simple form

$$\boldsymbol{\nabla}_\theta Q_P(x; \boldsymbol{\theta}) = \begin{bmatrix} \partial_{q_{min}} Q_U(x; \boldsymbol{\theta}) \\ \partial_{q_{max}} Q_U(x; \boldsymbol{\theta}) \end{bmatrix} = \begin{cases} [1, 0]^T & |x| \le q_{min} \\ [0, 0]^T & q_{min} < |x| \le q_{max} \\ [0, 1]^T & |x| > q_{max} \end{cases} , \tag{6}$$

which has again a bounded gradient magnitude and independent components and is, hence, best suited for first order gradient based optimization.

## 2.2 Constraints on $\theta$

In practice, for an efficient hardware implementation, we need to ensure that the quantization parameters only take specific discrete values: for uniform quantization, only integer values are allowed for the bitwidth $b$, and the stepsize $d$ must be a power-of-two, see e.g. (Jain et al., 2019); for power-of-two quantization, the bitwidth must be an integer, and the minimum and maximum absolute values $q_{min}$ and $q_{max}$ must be powers-of-two.

We fulfill these constraints by rounding the parameters in the forward pass to the closest integer or power-of-two value. In the backward pass we update the original float values, i.e., we used again the STE to propagate the gradients.

## 2.3 Experimental comparison of DQ parametrizations

In the following we compare the parametrizations using two experiments.

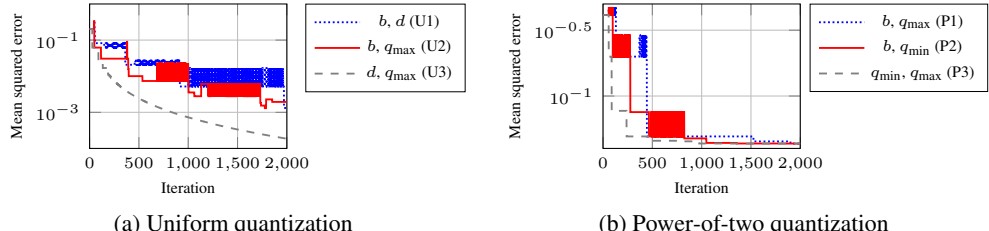

(a) Uniform quantization          (b) Power-of-two quantization

Figure 3: MSE for quantizing Gaussian data $x \sim N(0, 1)$ with uniform and power-of-two quantization. Parametrizations "U3" and "P3" converge to the lowest MSE without any oscillations.

*1) Quantization of Gaussian data*    In our first experiment we use DQ to learn the optimal quantization parameters $\boldsymbol{\theta}^*$ which minimize the *mean squared error* (MSE) $\mathrm{E}\left[\frac{1}{2}(Q(x; \boldsymbol{\theta}) - x)^2\right]$ with gradient descent and compare the convergence speed for three possible parametrizations of a uniform and power-of-two quantizer. We choose this example as the gradient $\boldsymbol{\nabla}_\theta Q(x; \boldsymbol{\theta}) = \mathrm{E}\left[(Q(x; \boldsymbol{\theta}) - x)\boldsymbol{\nabla}_\theta Q(x; \boldsymbol{\theta})\right]$ is just a scaled version of $\boldsymbol{\nabla}_\theta Q(x; \boldsymbol{\theta})$, i.e., the gradient direction depends directly on the parametrization of $Q(x; \boldsymbol{\theta})$ and thus the effects of changing the parametrization can be observed.

It is interesting to study the Hessian $\mathbf{H} = \boldsymbol{\nabla}_\theta \boldsymbol{\nabla}_\theta^T \mathrm{E}\left[(Q(x; \boldsymbol{\theta}) - x)^2\right] \in \mathbb{R}^{2 \times 2}$ of the MSE:

$$\mathbf{H} = \mathrm{E}\left[\boldsymbol{\nabla}_\theta Q(x; \boldsymbol{\theta})\boldsymbol{\nabla}_\theta Q(x; \boldsymbol{\theta})^T + (Q(x; \boldsymbol{\theta}) - x)\boldsymbol{\nabla}_\theta \boldsymbol{\nabla}_\theta^T Q(x; \boldsymbol{\theta})\right] \approx \mathrm{E}\left[\boldsymbol{\nabla}_\theta Q(x; \boldsymbol{\theta})\boldsymbol{\nabla}_\theta Q(x; \boldsymbol{\theta})^T\right].$$

Note that we use the outer-product approximation (Bishop, 2006) in order to simplify our considerations. From this equation it is apparent that the Hessian will be diagonal for the case U3 as $\boldsymbol{\nabla}_\theta Q(x; \boldsymbol{\theta})\boldsymbol{\nabla}_\theta Q(x; \boldsymbol{\theta})^T$ only contains an element in either $(1, 1)$ or $(2, 2)$ and, therefore, $\mathrm{E}\left[\boldsymbol{\nabla}_\theta Q(x; \boldsymbol{\theta})\boldsymbol{\nabla}_\theta Q(x; \boldsymbol{\theta})^T\right]$ is a diagonal matrix. From this, we can see that gradient descent with an individual learning rate for each parameter is equivalent to Newton's method and, therefore, efficient. In general this will not be the case for U1 and U2.

We conduct an experiment, using ADAM to optimize the mean squared quantization error on artificially generated data, which is generated by drawing $10^4$ samples from $N(0, 1)$. Please note that the same example was studied in (Jain et al., 2019). The results in Fig. 3 clearly show that the parametrizations "U3" and "P3" are best suited to optimize the uniform and power-of-two quantization parameters, respectively. Indeed, these quantizers converge without oscillation to the lowest MSE. It is interesting to see, that even adaptive gradient methods like ADAM can not solve the scaling issue described in Sec. 2.1. In the Appendix A.4 we give further empirical evidence to support this claim and compare the different parametrizations for the training of a quantized ResNet-20 on CIFAR-10 using ADAM. Note that all cases use the same learning rate. For the interested reader, a more detailed visualization of the error surfaces over the quantization parameters can be found in Appendix A.3.

*2) CIFAR-10*    In our second experiment we train a ResNet-20 (He et al., 2016) with quantized parameters and activations on CIFAR-10 (Krizhevsky & Hinton, 2009) using the same settings as proposed by (He et al., 2016). Fig. 4 shows the evolution of the training and validation error during training for the case of uniform quantization. The plots for power-of-two quantization can be found in the appendix (Fig. 10). We initialize this network from random parameters or from a pre-trained float network. The quantized DNNs are trained for 160 epochs, using SGD with momentum 0.9 and a learning rate schedule starting with 0.01 and reducing it by a factor of 10 after 80 and 120 epochs, respectively. We use random flips and crops for data augmentation. Each epoch takes about 2.5 min on a single GTX 1080 Ti.

In case of randomly initialized weights, we use an initial stepsize $d_l = 2^{-3}$ for the quantization of weights and activations. Otherwise, we initialize the weights using a pre-trained floating point network and the initial stepsize for a layer is chosen to be $d_l = 2^{\lfloor \log_2(\max |\boldsymbol{\mathcal{W}}_l|/(2^{b-1}-1)) \rfloor}$. The remaining quantization parameters are chosen such that we start from an initial bitwidth of $b = 4$ bit. This is a reasonable upper limit for $b$, as in practice no performance degradation can be observed for $b > 4$bit. Even simple offline algorithms like min/max quantization result in networks with good accuracies. We define no memory constraints during training, i.e., the network can learn to use a large number of bits to quantize weights and activations of each layer. From Fig. 4, we again observe that the parametrization $\boldsymbol{\theta} = [d, q_{\max}]^T$ is best suited to train a uniformly quantized DNN as it converges

Table 1: Comparison of different DQ parametrizations for ResNet-20 on CIFAR-10.
(validation error with "random"/"float net" initialization)

| Quantization | Float32 | Uniform quantization | | | Power-of-two quantization | | |
| --- | --- | --- | --- | --- | --- | --- | --- |
| | | $\boldsymbol{\theta} = [b, d]^T$ | $\boldsymbol{\theta} = [b, q_{\max}]^T$ | $\boldsymbol{\theta} = [d, q_{\max}]^T$ | $\boldsymbol{\theta} = [b, q_{\max}]^T$ | $\boldsymbol{\theta} = [b, q_{\min}]^T$ | $\boldsymbol{\theta} = [q_{\min}, q_{\max}]^T$ |
| Weights | 8.50%/7.29% | 17.8%/8.18% | 8.80%/7.44% | **8.50%/7.32%** | 11.70%/7.90% | 53.07%/23.01% | **10.61%/7.56%** |
| Weights+Activations | | 28.9%/9.03% | 9.43%/7.74% | **9.23%/7.40%** | 22.91%/11.68% | diverging/35.68% | **15.10%/9.86%** |

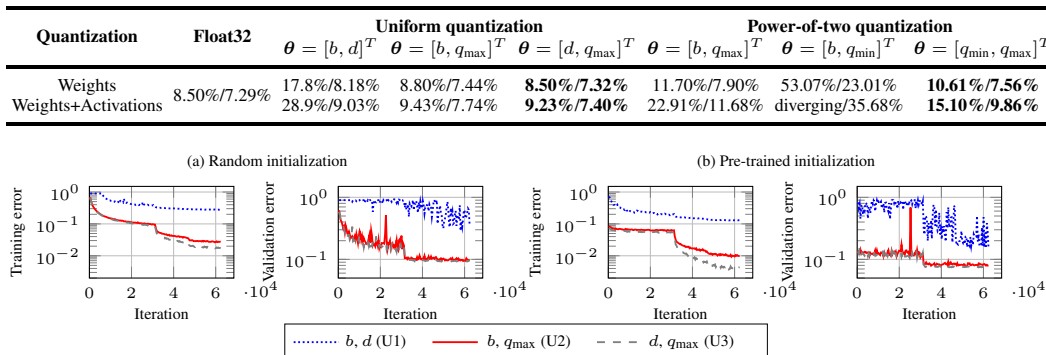

Figure 4: ResNet-20 with uniformly quantized weights and activations.

to the best local optimum. Furthermore, we observe the smallest oscillation of the validation error for this parametrization.

Table 1 compares the best validation error for all parametrizations of the uniform and power-of-two quantizations. We trained networks either with quantized weights and full precision activations or with both being quantized. In case of activation quantization with power-of-two, we use one bit to explicitly represent the value $x = 0$. This is advantageous as the ReLU nonlinearity will map many activations to this value. We can observe that training the quantized DNN with the optimal parametrization of DQ, i.e., using either $\boldsymbol{\theta} = [d, q_{\max}]^T$ or $\boldsymbol{\theta} = [q_{\min}, q_{\max}]^T$ results in a network with the lowest validation error. This result again supports our theoretical considerations from Sec. 2.1.

## 3 TRAINING QUANTIZED DNNS WITH MEMORY CONSTRAINTS

We now discuss how to train quantized DNNs with memory constraints. Such constraints appear in many applications when the network inference is performed on an embedded device with limited computational power and memory resources.

A quantized DNN consists of layers which compute

$$\boldsymbol{\mathcal{X}}_l = f_l(Q(\boldsymbol{\mathcal{W}}_l; \boldsymbol{\theta}_l^w) * Q(\boldsymbol{\mathcal{X}}_{l-1}; \boldsymbol{\theta}_{l-1}^x) + Q(\boldsymbol{c}_l; \boldsymbol{\theta}_l^w)) \quad \text{with} \quad l = 1, ..., L, \tag{7}$$

where $f_l(\cdot)$ denotes the nonlinear activation function of layer $l$ and $Q(\cdot; \boldsymbol{\theta})$ is a per-tensor quantization with parameters $\boldsymbol{\theta}$ applied separately to the input and output tensors $\boldsymbol{\mathcal{X}}_{l-1} \in \mathcal{I}_l$ and $\boldsymbol{\mathcal{X}}_l \in \mathcal{I}_l$, and also to both the weight tensors $\boldsymbol{\mathcal{W}}_l \in \mathcal{P}_l$ and the bias vector $\boldsymbol{c}_l \in \mathbb{R}^{M_l}$.[1] For a fully connected layer, $\mathcal{I}_{l-1} = \mathbb{R}^{M_{l-1}}$, $\mathcal{I}_l = \mathbb{R}^{M_l}$ are vectors, $\mathcal{P}_l = \mathbb{R}^{M_l \times M_{l-1}}$ are matrices and $A * B$ is a matrix-vector product. In case of a convolutional layer, $\mathcal{I}_{l-1} = \mathbb{R}^{M_{l-1} \times N_{l-1} \times N_{l-1}}$, $\mathcal{I}_l = \mathbb{R}^{M_l \times N_l \times N_l}$, $\mathcal{P}_l = \mathbb{R}^{M_l \times M_{l-1} \times K_l \times K_l}$ are tensors and $A * B$ is a set of $M_{l-1}M_l$ 2D convolutions, where the convolution is performed on square-sized feature maps of size $N_{l-1} \times N_{l-1}$ using square-sized kernels of size $K_l \times K_l$.

DNNs with quantized weights and activations have a smaller memory footprint and are also computationally cheaper to evaluate since $Q(\alpha; \boldsymbol{\theta}) \cdot Q(\beta; \boldsymbol{\theta})$ for $\alpha, \beta \in \mathbb{R}$ requires only an integer multiplication for the case of uniform quantization or an integer addition of the exponents for power-of-two quantization. Furthermore, $Q(\alpha; \boldsymbol{\theta}) + Q(\beta; \boldsymbol{\theta})$ for $\alpha, \beta \in \mathbb{R}$ only requires an integer addition. Table 2 compares the computational complexity and the memory footprint of layers which apply uniform or power-of-two quantization to weights and activations.

We consider the following memory characteristics of the DNN, constraining them during training:

1. *Total memory $S^w(\boldsymbol{\theta}_1^w, ..., \boldsymbol{\theta}_L^w) = \sum_{l=1}^L S_l^w(\boldsymbol{\theta}_l^w)$ to store all weights:* We use the constraint

$$g_1(\boldsymbol{\theta}_1^w, ..., \boldsymbol{\theta}_L^w) = S^w(\boldsymbol{\theta}_1^w, ..., \boldsymbol{\theta}_L^w) - S_0^w = \sum_{l=1}^L S_l^w(\boldsymbol{\theta}_l^w) - S_0^w \leq 0, \tag{8a}$$

---

[1] In this paper, we use "weights" to refer to $\boldsymbol{\mathcal{W}}$ and $\boldsymbol{c}$.

Table 2: Number of multiplications $C_l^{mul}$, additions $C_l^{add}$ as well as required memory to store the weights $S_l^w$ and activations $S_l^x$ of fully connected and convolutional layers.

| Layer | Quantization | $C_l^{mul}$ | $C_l^{add}$ | $S_l^w$ | $S_l^x$ |
|---|---|---|---|---|---|
| Fully connected | uniform | $M_l M_{l-1}$ | $M_l M_{l-1}$ | $M_l(M_{l-1}+1)b_l^w$ | $M_l b_l^x$ |
| | pow-2 | $0$ | $2M_l M_{l-1}$ | | |
| Convolutional | uniform | $M_l M_{l-1} N_l^2 K_l^2$ | $M_l M_{l-1} N_l^2 K_l^2$ | $M_l(M_{l-1}K_l^2+1)b_l^w$ | $M_l N_l^2 b_l^x$ |
| | pow-2 | $0$ | $2M_l M_{l-1} N_l^2 K_l^2$ | | |

to ensure that the total weight memory requirement $S^w(\boldsymbol{\theta}_1^w, ..., \boldsymbol{\theta}_L^w)$ is smaller than a certain maximum weight memory size $S_0^w$. Table 2 gives $S_l^w(\boldsymbol{\theta}_l^w)$ for the case of fully connected and convolutional layers. Each layer's memory requirement $S_l^w(\boldsymbol{\theta}_l^w)$ depends on the bitwidth $b_l^w$: reducing $S_l^w(\boldsymbol{\theta}_l^w)$ will reduce the bitwidth $b_l^w$.

2. *Total activation memory* $S^x(\boldsymbol{\theta}_1^x, ..., \boldsymbol{\theta}_L^x) = \sum_{l=1}^L S_l^x(\boldsymbol{\theta}_l^x)$ *to store all feature maps:* We use the constraint

$$g_2(\boldsymbol{\theta}_1^x, ..., \boldsymbol{\theta}_L^x) = S^x(\boldsymbol{\theta}_1^x, ..., \boldsymbol{\theta}_L^x) - S_0^x = \sum_{l=1}^L S_l^x(\boldsymbol{\theta}_l^x) - S_0^x \leq 0, \tag{8b}$$

to ensure an upper limit on the total activation memory size $S_0^x$. Table 2 gives $S_l^x(\boldsymbol{\theta}_l^x)$ for the case of fully connected and convolutional layers. Such a constraint is important if we use pipelining for accelerated inference, i.e., if we evaluate multiple layers with several consecutive inputs in parallel. This can, e.g., be the case for FPGA implementations (Guo et al., 2017).

3. *Maximum activation memory* $\hat{S}^x(\boldsymbol{\theta}_1^x, ..., \boldsymbol{\theta}_L^x) = \max_{l=1,...,L} S_l^x$ *to store the largest feature map:* We use the constraint

$$g_3(\boldsymbol{\theta}_1^x, ..., \boldsymbol{\theta}_L^x) = \hat{S}^x(\boldsymbol{\theta}_1^x, ..., \boldsymbol{\theta}_L^x) - \hat{S}_0^x = \max_{l=1,...,L}(S_l^x) - \hat{S}_0^x \leq 0, \tag{8c}$$

to ensure that the maximum activation size $\hat{S}^x$ does not exceed a given limit $\hat{S}_0^x$. This constraint is relevant for DNN implementations where layers are processed sequentially.

To train the quantized DNN with memory constraints, we need to solve the optimization problem

$$\min_{\boldsymbol{\mathcal{W}}_l, \boldsymbol{c}_l, \boldsymbol{\theta}_l^w, \boldsymbol{\theta}_l^x} \mathrm{E}_{p(\boldsymbol{x}, \boldsymbol{y})}\left[J(\boldsymbol{\mathcal{X}}_L, \boldsymbol{\mathcal{Y}})\right] \quad \text{s.t.} \quad g_j(\boldsymbol{\theta}_1^w, ..., \boldsymbol{\theta}_L^w, \boldsymbol{\theta}_1^x, ..., \boldsymbol{\theta}_L^x) \leq 0 \quad \text{for all } j = 1, ..., 3 \tag{9}$$

where $J(\boldsymbol{\mathcal{X}}_L, \boldsymbol{\mathcal{Y}})$ is the loss function for yielding the DNN output $\boldsymbol{\mathcal{X}}_L$ although the ground truth is $\boldsymbol{\mathcal{Y}}$. Eq. (9) learns the weights $\boldsymbol{\mathcal{W}}_l$, $\boldsymbol{c}_l$ as well as the quantization parameters $\boldsymbol{\theta}_l^x$, $\boldsymbol{\theta}_l^w$. In order to use simple stochastic gradient descent solvers, we use the penalty method (Bertsekas, 2014) to convert (9) into the unconstrained optimization problem

$$\min_{\boldsymbol{\mathcal{W}}_l, \boldsymbol{c}_l, \boldsymbol{\theta}_l^w, \boldsymbol{\theta}_l^x} \mathrm{E}_{p(\boldsymbol{x}, \boldsymbol{y})}\left[J(\boldsymbol{\mathcal{X}}_L, \boldsymbol{\mathcal{Y}})\right] + \sum_{j=1}^J \lambda_j \max(0, g_j(\boldsymbol{\theta}_1^w, ..., \boldsymbol{\theta}_L^w, \boldsymbol{\theta}_1^x, ..., \boldsymbol{\theta}_L^x))^2, \tag{10}$$

where $\lambda_j \in \mathbb{R}^+$ are individual weightings for the penalty terms. Hence, training with weight and activation size constraints requires choosing two penalty weightings $\lambda_j$, one for (8a) and one for either (8b) or (8c).

Note, that the optimization problem (10) does not necessarily give a quantized DNN which fulfills the memory constraints. The probability to fulfill the constraint $g_j$ depends on the choice of $\lambda_j$. In particular, this probability increases with larger $\lambda_j$. However, choosing a too large $\lambda_j$ will yield a penalty term that dominates over the network loss decreasing the network performance. In our experiments, we choose $\lambda_j$ such that the initial loss and the penalty term have approximately the same magnitude. Using this simple heuristic, we optained quantized DNNs that reached a high accuracy and at the same time fulfilled the constraints at the end of training.

## 4 EXPERIMENTS

In the following, we will use the best parametrizations for uniform and power-of-two DQ, i.e., $\boldsymbol{\theta}_U = [d, q_{\max}]^T$ and $\boldsymbol{\theta}_P = [q_{\min}, q_{\max}]^T$, that we found in Sec. 2. Both parametrizations do not directly depend on the bitwidth $b$. Therefore, we compute it by using $b(\boldsymbol{\theta}_U) = \left\lceil \log_2\left(\frac{q_{\max}}{d} + 1\right) + 1 \right\rceil$ and $b(\boldsymbol{\theta}_P) = \left\lceil \log_2\left(\log_2\left(\frac{q_{\max}}{q_{\min}}\right) + 1\right) + 1 \right\rceil$. All quantized networks use a pre-trained float32 network for initialization and all quantizers are initialized as described in Sec. 2.3. For our experiments on CIFAR-10, we use the same training setup as described in Sec. 2.3. For the experiments on

Table 3: Homogeneous vs. heterogeneous quantization of ResNet-20 on CIFAR-10.

| | Bitwidth Weight/Activ. | $q_{max}$ Weight/Activ. | Size Weight/Activ.(max)/Activ.(sum) | Uniform quant. Validation error | Power-of-two quant. Validation error |
|---|---|---|---|---|---|
| Baseline | 32bit/32bit | – | 1048KB/64KB/736KB | 7.29% | |
| Fixed | 2bit/32bit | fixed/– | 65.5KB/64KB/736KB | 10.81% | 8.99% |
| TQT (Jain et al., 2019) | 2bit/32bit | learned/ – | 65.5KB/64KB/736KB | 9.47% | 8.79% |
| Ours (w/ constr. (8a)) | learned/32bit | learned/- | 70KB/64KB/736KB | 8.59% | 8.53% |
| Fixed | 2bit/4bit | fixed/fixed | 65.5KB/8KB/92KB | 11.30% | 11.62% |
| TQT (Jain et al., 2019) | 2bit/4bit | learned/learned | 65.5KB/8KB/92KB | 9.62% | 11.29% |
| Ours (w/ constr. (8a) and (8b)) | learned/learned | learned/learned | 70KB/ – /92KB | 9.38% | 11.29% |
| Ours (w/ constr. (8a) and (8c)) | learned/learned | learned/learned | 70KB/8KB/ – | 8.58% | 11.23% |

ImageNet, we train the quantized DNNs for 50 epochs, using SGD with momentum 0.9 and a learning rate schedule starting with 0.01 and reducing it by a factor of 10 after 16 and 32 epochs, respectively. Please note that we quantize *all* layers opposed to other papers which use a higher precision for the first and/or last layer.

In our experiments, we noticed that the performance of DQ is not sensitive to the choice of $\lambda_j$ in (10). For the CIFAR-10 experiments, we use $\lambda = 0.1$ for both constraints (for sizes in kB). For the ImageNet experiments, we kept the same regularization level by scaling $\lambda_j$ with the square of the size ratio between the ImageNet model and the CIFAR-10 model. We scale with the square-ratio as the constraints in (10) are squared penalty terms.

First, in Table 3/top, we train a ResNet-20 on CIFAR-10 with quantized weights and float32 activations. We start with the most restrictive quantization scheme with fixed $q_{max}$ and $b = 2$bit ("Fixed"). Then, we allow the model to learn $q_{max}$ while $b = 2$bit remains fixed as was done in (Jain et al., 2019) ("TQT"). Finally, we learn both $q_{max}$ and $b$ with the constraint that the weight size is at most 70KB ("Ours"), which is just 4.5kB larger that the previous 2Bit networks. This allows the model to allocate more than two bits to some layers. From Table 3/top, we observe that the error is smallest when we learn all quantization parameters.

In Table 3/bottom, weights and activations are quantized. For activation quantization, we consider two cases as discussed in Sec. 3. The first one constrains the total activation memory $S^x$ while the second constrains the maximum activation memory $\hat{S}^x$ such that both have the same size as a homogeneously quantized model with 4bit activations. Again, we observe that the error is smallest when we learn all quantization parameters.

We also use DQ to train quantized ResNet-18 (He et al., 2016) and MobileNetV2 (Sandler et al., 2018) on ImageNet (Deng et al., 2009) with 4bit uniform weights and activations or equivalent-sized networks with learned quantization parameters. This is quite aggressive and, thus, a fixed quantization scheme loses more than 6% accuracy while our quantization scheme loses less than 0.5% compared to a float32 precision network.

Our results compare favorably to other recent quantization approaches. To our knowledge, the best result for a 4bit ResNet-18 was reported by (Esser et al., 2019) (29.91% error). This is very close to our performance (29.92% error). Importantly, (Esser et al., 2019) did not quantize the first and last layers, meaning that their network is much bigger. Specifically, compared to our quantized ResNet-18, their model with high precision input and output layers requires 37% more memory to store the weights. Moreover, (Esser et al., 2019) learns stepsizes which are not restricted to powers-of-two. As explained in Sec. 2.2, uniform quantization with power-of-two stepsize leads to more efficient inference, effectively allowing to efficiently compute any multiplication with an integer multiplication and bit-shift. To our knowledge only (Wang et al., 2018) reported results of MobileNetV2 quantized to 4bit. They keep the baseline performance constraining the network to the same size as the 4bit network. However, they do not quantize the activations in this case. In addition, DQ training is efficient since it is comparable to the training of unquantized network. Specifically, one epoch on ImageNet takes 37min for MobileNetV2 and 18min for ResNet-18 on four Nvidia Tesla V100.

Fig. 5 shows the weight bitwidth assignment over layers. We observe that small bitwidths are used for layers with many parameters, i.e., pointwise convolutions and fully connected layers. However, the resulting bitwidth assignments are complex, meaning that there is no simple heuristic. Therefore, it is important to learn the optimal bitwidth assignment.

Table 4: Homogeneous vs. heterogeneous quantization of MobileNetV2 and ResNet-18 on ImageNet.

| | Bitwidth Weight/Activ. | $q_{max}$ Weight/Activ. | MobileNetV2 Size Weight/Activ(max) | Validation Error | ResNet-18 Size Weight/Activ(max) | Validation Error |
|---|---|---|---|---|---|---|
| Baseline | 32bit/32bit | – | 13.23MB/4.59MB | 29.82% | 44.56MB/3.04MB | 29.72% |
| Fixed | 4bit/4bit | fixed/fixed | 1.65MB/0.57MB | 36.27% | 5.57MB/0.38MB | 34.15% |
| TQT (Jain et al., 2019) | 4bit/4bit | learned/learned | 1.65MB/0.57MB | 32.21% | 5.57MB/0.38MB | 30.49% |
| Ours (w/ constr. (8a) and (8c)) | learned/learned | learned/learned | 1.55MB/0.57MB | 30.26% | 5.40MB/0.38MB | 29.92% |
| Ours (w/o constr.) | learned/learned | learned/learned | 3.14MB/1.58MB | 29.41% | 10.50MB/1.05MB | 29.34% |

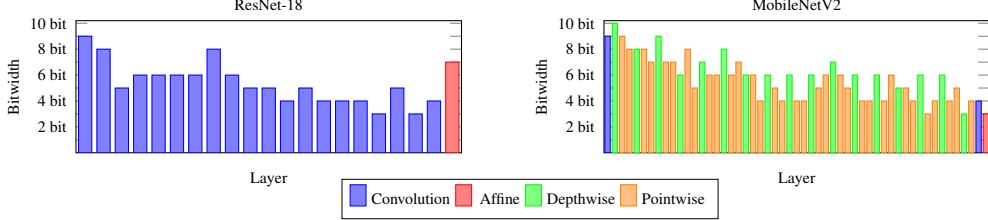

Figure 5: Weight bitwidth assignment over layers for ResNet-18 and MobileNetV2 on ImageNet with weights constrained to a maximum size of 5.57MB. Our method has learned a heterogeneous bitwidth distribution, which gives a better performance than a homogeneous one (see Table 4).

## 5 CONCLUSIONS

In this paper we discussed differentiable quantization and its application to the training of compact DNNs with memory constraints. In order to fulfill memory constraints, we introduced penalty functions during training and used stochastic gradient descent to find the optimal weights as well as the optimal quantization values in a joint fashion. We showed that there are several possible parametrizations of the quantization function. In particular, learning the bitwidth directly is not optimal; therefore, we proposed to parametrize the quantizer with the stepsize and dynamic range instead. The bitwidth can then be inferred from them. This approach is competitive to other recent quantization methods while it does not require to retrain the network multiple times in contrast to reinforcement learning approaches (Wang et al., 2018; Elthakeb et al., 2018).

### ACKNOWLEDGEMENTS

We would like to thank Masato Ishii for many helpful comments during the preparation of this manuscript.

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

# A   DERIVATION OF THE GRADIENTS FOR DIFFERENTIABLE QUANTIZATION (DQ)

In the following sections, we will give the derivatives $\frac{\partial}{\partial x}Q(x;\boldsymbol{\theta})$ and gradients $\nabla_{\boldsymbol{\theta}}Q(x;\boldsymbol{\theta})$ for the uniform and the power-of-two quantizers. The results are summarized in Sec. 2.

We use the straight-through gradient estimate whenever we need to differentiate a non-differentiable floor function, i.e., we assume

$$\frac{\partial}{\partial x}\lfloor x \rfloor = 1. \tag{11}$$

## A.1   DERIVATIVES OF THE UNIFORM QUANTIZER

Fig. 6(a) shows a symmetric uniform quantizer $Q_U(x;\boldsymbol{\theta})$ which maps a real value $x \in \mathbb{R}$ to one of $I = 2k + 1$ quantized values $q \in \{-kd, ..., 0, ..., kd\}$ by computing

$$q = Q_U(x;\boldsymbol{\theta}) = \text{sign}(x) \begin{cases} d\left\lfloor \frac{|x|}{d} + \frac{1}{2} \right\rfloor & |x| \leq q_{\max} \\ q_{\max} & |x| > q_{\max} \end{cases} \tag{12}$$

using the parameters $\boldsymbol{\theta} = [d, q_{\max}, b]^T$ where $d \in \mathbb{R}$ is the stepsize, $q_{\max} \in \mathbb{R}$ is the maximum value and $b \in \mathbb{N}$ is the number of bits that we use to encode the quantized values $q$. The elements of $\boldsymbol{\theta}$ are dependent as there is the relationship $q_{\max} = (2^{b-1} - 1)d$.

### A.1.1   CASE U1: PARAMETRIZATION WITH RESPECT TO $b$ AND $d$

For the parametrization with respect to the bitwidth $b$ and steps size $d$, (12) is given by

$$q = Q_U(x;b,d) = \text{sign}(x)d \begin{cases} \left\lfloor \frac{|x|}{d} + \frac{1}{2} \right\rfloor & |x| \leq (2^{b-1} - 1)d \\ 2^{b-1} - 1 & |x| > (2^{b-1} - 1)d \end{cases} \tag{13}$$

and the derivatives are given by

$$\frac{\partial Q_U(x;b,d)}{\partial b} = \text{sign}(x)\frac{2^{b-1}\log 2}{2^{b-1} - 1} \begin{cases} 0 & |x| \leq (2^{b-1} - 1)d \\ (2^{b-1} - 1)d & |x| > (2^{b-1} - 1)d \end{cases}, \tag{14a}$$

$$\frac{\partial Q_U(x;b,d)}{\partial d} = \text{sign}(x)\frac{1}{d} \begin{cases} d\left\lfloor \frac{|x|}{d} + \frac{1}{2} \right\rfloor - |x| & |x| \leq (2^{b-1} - 1)d \\ (2^{b-1} - 1)d & |x| > (2^{b-1} - 1)d \end{cases}. \tag{14b}$$

### A.1.2   CASE U2: PARAMETRIZATION WITH RESPECT TO $b$ AND $q_{\text{MAX}}$

For the parametrization with respect to the bitwidth $b$ and maximum value $q_{\max}$, (12) is given by

$$q = Q_U(x;b,q_{\max}) = \text{sign}(x)q_{\max} \begin{cases} \frac{1}{2^{b-1}-1}\left\lfloor |x|\frac{2^{b-1}-1}{q_{\max}} + \frac{1}{2} \right\rfloor & |x| \leq q_{\max} \\ 1 & |x| > q_{\max} \end{cases} \tag{15}$$

and the derivatives are given by

$$\frac{\partial Q_U(x;b,q_{\max})}{\partial b} = \text{sign}(x)\frac{2^{b-1}\log 2}{2^{b-1} - 1} \begin{cases} -\frac{q_{\max}}{2^{b-1}-1}\left\lfloor |x|\frac{2^{b-1}-1}{q_{\max}} + \frac{1}{2} \right\rfloor + \beta_1 & |x| \leq q_{\max} \\ 0 & |x| > q_{\max} \end{cases}, \tag{16a}$$

$$\frac{\partial Q_U(x;b,q_{\max})}{\partial q_{\max}} = \text{sign}(x)\frac{1}{q_{\max}} \begin{cases} \frac{q_{\max}}{2^{b-1}-1}\left\lfloor |x|\frac{2^{b-1}-1}{q_{\max}} + \frac{1}{2} \right\rfloor + \beta_2 & |x| \leq q_{\max} \\ q_{\max} & |x| > q_{\max} \end{cases}, \tag{16b}$$

where $\beta_1 = \frac{q_{\max}}{2^{b-1}\log 2}\frac{\partial \left\lfloor |x|\frac{2^{b-1}-1}{q_{\max}} + \frac{1}{2} \right\rfloor}{\partial b} = |x|$ and $\beta_2 = \frac{q_{\max}^2}{2^{b-1}-1}\frac{\partial \left\lfloor |x|\frac{2^{b-1}-1}{q_{\max}} + \frac{1}{2} \right\rfloor}{\partial q_{\max}} = -|x|$, if we use the straight-through gradient estimate for the floor function.

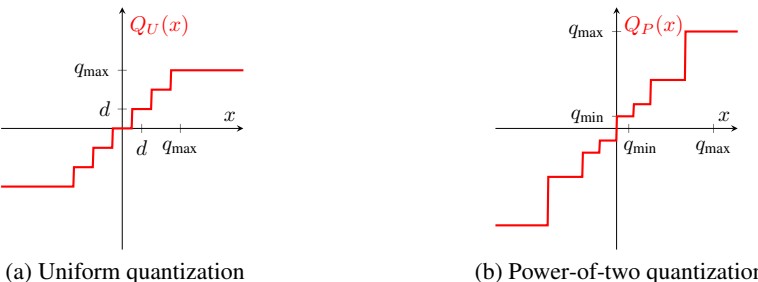

(a) Uniform quantization                    (b) Power-of-two quantization

Figure 6: Examples of uniform quantizer $Q_U(x)$ and power-of-two quantizer $Q_P(x)$ for $b = 3$ bits

### A.1.3 CASE U3: PARAMETRIZATION WITH RESPECT TO $d$ AND $q_{MAX}$

Eq. (12) gives the quantization with respect to the step size $d$ and maximum value $q_{max}$. The derivatives are

$$\frac{\partial Q_U(x; d, q_{max})}{\partial d} = \text{sign}(x)\frac{1}{d}\begin{cases} d\left\lfloor \frac{|x|}{d} + \frac{1}{2} \right\rfloor - |x| & |x| \leq q_{max} \\ 0 & |x| > q_{max} \end{cases}, \tag{17a}$$

$$\frac{\partial Q_U(x; d, q_{max})}{\partial q_{max}} = \text{sign}(x)\frac{1}{q_{max}}\begin{cases} 0 & |x| \leq q_{max} \\ q_{max} & |x| > q_{max} \end{cases}. \tag{17b}$$

### A.2 DERIVATIVES OF THE POWER-OF-TWO QUANTIZER

Power-of-two quantization $Q_P(x; \boldsymbol{\theta})$ maps a real-valued number $x \in \mathbb{R}$ to a quantized value $q \in \{\pm 2^k : k \in \mathbb{Z}\}$ by

$$q = Q_P(x; \boldsymbol{\theta}) = \text{sign}(x)\begin{cases} q_{min} & |x| \leq q_{min} \\ 2^{\lfloor 0.5 + \log_2 |x| \rfloor} & q_{min} < |x| \leq q_{max} \\ q_{max} & |x| > q_{max} \end{cases}, \tag{18}$$

where $q_{min}$ and $q_{max}$ are the minimum and maximum (absolute) values of the quantizer for a bitwidth of $b$ bits. Fig. 6b shows the quantization curve for this quantization scheme.

Using the STE for the floor operation, the derivative $\partial_x Q_P(x; \boldsymbol{\theta})$ is given by

$$\partial_x Q_P(x) = \begin{cases} 0 & |x| \leq q_{min} \\ \frac{2^{\lfloor 0.5 + \log_2 |x| \rfloor}}{|x|} & q_{min} < |x| \leq q_{max} \\ 0 & |x| > q_{max} \end{cases}. \tag{19}$$

The power-of-two quantization has the three parameters $\boldsymbol{\theta} = [b, q_{min}, q_{max}]$, which are dependent on each other, i.e., $q_{max} = 2^{2^{b-1}-1}q_{min}$. Therefore, we have again three different parametrizations with $\boldsymbol{\theta} = [b, q_{min}]$, $\boldsymbol{\theta} = [b, q_{max}]$ or $\boldsymbol{\theta} = [q_{min}, q_{max}]$, respectively. The resulting partial derivatives for each parametrization are shown in Fig. 7 and summarized in the following sections. Similar to the uniform case, one parametrization ($\boldsymbol{\theta} = [q_{min}, q_{max}]$) leads to a gradient with the nice form

$$\boldsymbol{\nabla}_\theta Q_P(x; \boldsymbol{\theta}) = \begin{bmatrix} \partial_{q_{min}} Q_U(x; \boldsymbol{\theta}) \\ \partial_{q_{max}} Q_U(x; \boldsymbol{\theta}) \end{bmatrix} = \begin{cases} [1, 0]^T & |x| \leq q_{min} \\ [0, 0]^T & q_{min} < |x| \leq q_{max} \\ [0, 1]^T & |x| > q_{max} \end{cases}, \tag{20}$$

which has a bounded gradient magnitude and independent components and is, hence, well suited for first order gradient based optimization.

### A.2.1 CASE P1: PARAMETRIZATION WITH RESPECT TO $b$ AND $q_{MAX}$

For the parametrization with $\boldsymbol{\theta} = [b, q_{max}]$, (18) is given by

$$Q_P(x; b, q_{max}) = \text{sign}(x)\begin{cases} 2^{-2^{b-1}+1}q_{max} & |x| \leq 2^{-2^{b-1}+1}q_{max} \\ 2^{\lfloor 0.5 + \log_2 |x| \rfloor} & 2^{-2^{b-1}+1}q_{max} < |x| \leq q_{max} \\ q_{max} & |x| > q_{max} \end{cases} \tag{21}$$

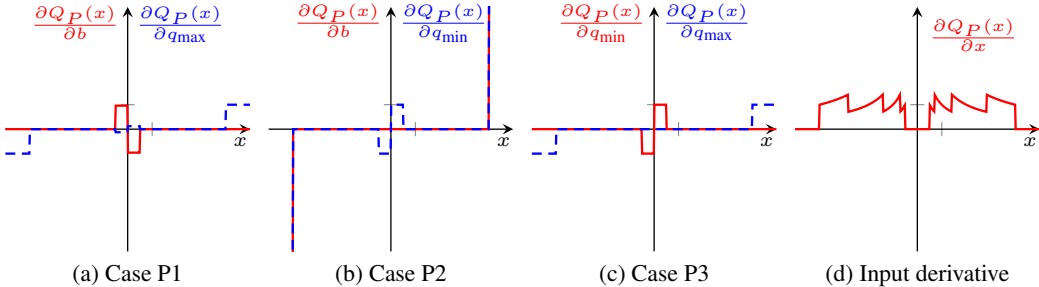

$$\text{(a) Case P1} \qquad \text{(b) Case P2} \qquad \text{(c) Case P3} \qquad \text{(d) Input derivative}$$

Figure 7: Derivatives for the three different parametrizations of $Q_P(x; \boldsymbol{\theta})$

and the partial derivatives are

$$\frac{\partial Q_P(x; b, q_{\max})}{\partial b} = \text{sign}(x) \begin{cases} -2^{-2^{b-1}+b}(\log 2)^2 q_{\max} & |x| \leq -2^{-2^{b-1}+1}q_{\max} \\ 0 & -2^{-2^{b-1}+1}q_{\max} < |x| \leq q_{\max} \\ 0 & |x| > q_{\max} \end{cases}, \tag{22a}$$

$$\frac{\partial Q_P(x; b, q_{\max})}{\partial q_{\max}} = \text{sign}(x) \begin{cases} 2^{-2^{b-1}+1} & |x| \leq -2^{-2^{b-1}+1}q_{\max} \\ 0 & -2^{-2^{b-1}+1}q_{\max} < |x| \leq q_{\max} \\ 1 & |x| > q_{\max} \end{cases}. \tag{22b}$$

### A.2.2 CASE P2: PARAMETRIZATION WITH RESPECT TO $b$ AND $q_{\text{MIN}}$

For the parametrization with $\boldsymbol{\theta} = [b, q_{\min}]$, (18) is given by

$$Q_P(x; b, q_{\min}) = \text{sign}(x) \begin{cases} q_{\min} & |x| \leq q_{\min} \\ 2^{\lfloor 0.5+\log_2 |x| \rfloor} & q_{\min} < |x| \leq 2^{2^{b-1}-1}q_{\min} \\ 2^{2^{b-1}-1}q_{\min} & |x| > 2^{2^{b-1}-1}q_{\min} \end{cases} \tag{23}$$

and the partial derivatives are

$$\frac{\partial Q_P(x; b, q_{\min})}{\partial b} = \text{sign}(x) \begin{cases} 0 & |x| \leq q_{\min} \\ 0 & q_{\min} < |x| \leq 2^{2^{b-1}-1}q_{\min} \\ 2^{2^{b-1}+b-2}(\log 2)^2 q_{\min} & |x| > 2^{2^{b-1}-1}q_{\min} \end{cases}, \tag{24a}$$

$$\frac{\partial Q_P(x; b, q_{\min})}{\partial q_{\min}} = \text{sign}(x) \begin{cases} 1 & |x| \leq q_{\min} \\ 0 & q_{\min} < |x| \leq 2^{2^{b-1}-1}q_{\min} \\ 2^{2^{b-1}-1} & |x| > 2^{2^{b-1}-1}q_{\min} \end{cases}. \tag{24b}$$

### A.2.3 CASE P3: PARAMETRIZATION WITH RESPECT TO $q_{\text{MIN}}$ AND $q_{\text{MAX}}$

Eq. (18) gives the parametrization of $Q(x; \boldsymbol{\theta})$ with respect to the minimum value $q_{\min}$ and maximum value $q_{\max}$. The derivatives are

$$\frac{\partial Q_P(x; q_{\min}, q_{\max})}{\partial q_{\min}} = \text{sign}(x) \begin{cases} 1 & |x| \leq q_{\min} \\ 0 & q_{\min} < |x| \leq q_{\max} \\ 0 & |x| > q_{\max} \end{cases}, \tag{25a}$$

$$\frac{\partial Q_P(x; q_{\min}, q_{\max})}{\partial q_{\max}} = \text{sign}(x) \begin{cases} 0 & |x| \leq q_{\min} \\ 0 & q_{\min} < |x| \leq q_{\max} \\ 1 & |x| > q_{\max} \end{cases}. \tag{25b}$$

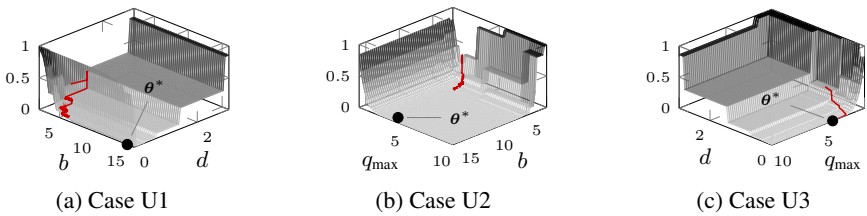

Figure 8: MSE surfaces for uniform quantization. Only U3 reaches the optimum $\boldsymbol{\theta}^*$.

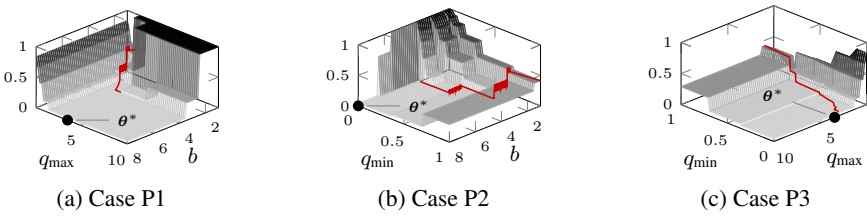

Figure 9: MSE surfaces for power-of-two quantization. Only P3 reaches the optimum $\boldsymbol{\theta}^*$.

### A.3 VISUALIZATION OF THE ERROR SURFACE FOR THE QUANTIZATION OF GAUSSIAN DATA

In Sec. 2.3 of the paper, we compared the three different parametrizations of the uniform quantizer at the example of optimal quantization of Gaussian data. To get a better understanding of Fig 3, we show how the error surfaces look like for this example problem. The experimental setup is the same as in Sec. 2.3, i.e., we use DQ to learn the optimal quantization parameters of a *uniform* and a *power-of-two* quantizer, which minimize the expected quantization error $\min_{\boldsymbol{\theta}} \mathrm{E}\left[(x - Q(x; \boldsymbol{\theta}))^2\right]$. We use three different parametrizations, adapt the quantizer's parameters with gradient descent and compare the convergence speed as well as the final quantization error. As an input, we generate $10^4$ samples from $N(0,1)$.

Fig. 8 shows the corresponding error surfaces for the three different parametrizations of the uniform quantization. The red curve shows the path through the parameter space taken by gradient descent in order to optimize the MSE, starting with the initial values $b = 2$, $d = q_{\max} = 1$. The optimum $\boldsymbol{\theta}^*$ is located at $b = 16$, $d \lessapprox 2^{-13}$, $q_{\max} = 4$, since we allow a maximal bitwidth of 16bit and the largest sample magnitude in our dataset is $\max\{x_1, ..., x_N\} \lessgtr 4$. In each of the cases U1-U3, the error surface is composed of steep ridges and large flat regions. The steep ridges force us to use small learning rates to avoid divergence. For cases U1 and U2, the optimal $\boldsymbol{\theta}^*$ can not be reached. However, for U3, $\boldsymbol{\theta}^*$ lies at the border of a flat region and can be easily reached. Furthermore, case U3 shows a much faster and more stable convergence without oscillation, since the gradient magnitudes are bounded and the error surface has fewer steep ridges where gradient descent starts oscillating.

Fig. 9 shows the corresponding error surfaces for the three different parametrizations of the power-of-two quantization. Again, the optimum $\boldsymbol{\theta}^*$ is not attained for two parametrizations, namely P1 and P2, as $\boldsymbol{\theta}^*$ is surrounded by a large, mostly flat region. For these two cases, gradient descent tends to oscillate at steep ridges and tends to be unstable. However, gradient descent converges to a point close to $\boldsymbol{\theta}^*$ for parametrization P3, where $\boldsymbol{\theta} = [q_{\min}, q_{\max}]$.

Finally, we also did a comparison of the different power-of-two quantizations on CIFAR-10. Fig. 10 shows the evolution of the training and validation error if we start from a random or a pre-trained float network initialization. We can observe that $\boldsymbol{\theta} = [q_{\min}, q_{\max}]$ has the best convergence behavior and thus also results in the smallest validation error (cf. Table 1). The unstable behavior of P2 is expected as the derivative $\frac{\partial Q_P}{\partial q_{\min}}$ can take very large (absolute) values.

### A.4 FURTHER EXPERIMENTS WITH ADAM

Finally, we did an experiment to verify that the parametrization is important, even if adaptive gradient descent methods like ADAM are used for optimization. Table 5 gives the results for a ResNet-20 trained on CIFAR-10. We observe, that again U3 and P3 are the best parametrizations.

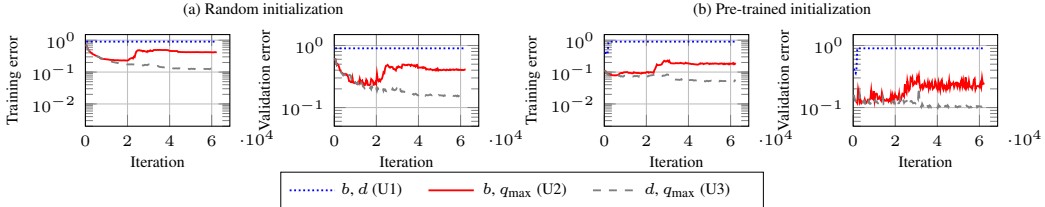

Figure 10: ResNet-20 with power-of-two quantized weights and activations.

Table 5: Error rate of ResNet-20 on CIFAR-10 using different quantization parametrizations. Training is done either by SGD with momentum or ADAM.

| Parametrization | SGD momentum | ADAM |
|---|---|---|
| U1 | 11.74 | 7.61 |
| U2 | 7.44 | 7.85 |
| U3 | **7.32** | **7.36** |
| P1 | 15.35 | 7.54 |
| P2 | 7.74 | 7.79 |
| P3 | **7.40** | **7.40** |

## B IMPLEMENTATION DETAILS FOR DIFFERENTIABLE QUANTIZATION

The following code gives our differentiable quantizer implementation in NNabla (Sony). The source code for reproducing our results will be published after the review process has been finished.

## B.1 UNIFORM QUANTIZATION

### B.1.1 CASE U1: PARAMETRIZATION WITH RESPECT TO $b$ AND $d$

```
1  def parametric_fixed_point_quantize_d_b(x, sign,
2                                          n_init, n_min, n_max,
3                                          d_init, d_min, d_max,
4                                          fix_parameters=False):
5      """Parametric version of `fixed_point_quantize` where the
6      bitwidth `b` and stepsize `d` are learnable parameters.
7
8      Returns:
9          ~nnabla.Variable: N-D array.
10     """
11     def clip_scalar(v, min_value, max_value):
12         return F.minimum_scalar(F.maximum_scalar(v, min_value), max_value)
13
14     def broadcast_scalar(v, shape):
15         return F.broadcast(F.reshape(v, (1,) * len(shape), inplace=False), shape=shape)
16
17     def quantize_pow2(v):
18         return 2 ** F.round(F.log(v) / np.log(2.))
19
20     n = get_parameter_or_create("n", (),
21                                 ConstantInitializer(n_init),
22                                 need_grad=True,
23                                 as_need_grad=not fix_parameters)
24     d = get_parameter_or_create("d", (),
25                                 ConstantInitializer(d_init),
26                                 need_grad=True,
27                                 as_need_grad=not fix_parameters)
28
29     # ensure that bitwidth is in specified range and an integer
30     n = F.round(clip_scalar(n, n_min, n_max))
31     if sign:
32         n = n - 1
33
34     # ensure that stepsize is in specified range and a power of two
35     d = quantize_pow2(clip_scalar(d, d_min, d_max))
36
37     # ensure that dynamic range is in specified range
38     xmax = d * (2 ** n - 1)
39
40     # compute min/max value that we can represent
41     if sign:
42         xmin = -xmax
43     else:
44         xmin = nn.Variable((1,), need_grad=False)
45         xmin.d = 0.
46
47     # broadcast variables to correct size
48     d = broadcast_scalar(d, shape=x.shape)
49     xmin = broadcast_scalar(xmin, shape=x.shape)
50     xmax = broadcast_scalar(xmax, shape=x.shape)
51
52     # apply fixed-point quantization
53     return d * F.round(F.clip_by_value(x, xmin, xmax) / d)
```

### B.1.2   CASE U2: PARAMETRIZATION WITH RESPECT TO $b$ AND $q_{\text{MAX}}$

```python
def parametric_fixed_point_quantize_b_xmax(x, sign,
                                           n_init, n_min, n_max,
                                           xmax_init, xmax_min, xmax_max,
                                           fix_parameters=False):
    """Parametric version of 'fixed_point_quantize' where the
    bitwidth 'b' and dynamic range 'xmax' are learnable parameters.

    Returns:
        ~nnabla.Variable: N-D array.
    """
    def clip_scalar(v, min_value, max_value):
        return F.minimum_scalar(F.maximum_scalar(v, min_value), max_value)

    def broadcast_scalar(v, shape):
        return F.broadcast(F.reshape(v, (1,) * len(shape), inplace=False), shape=shape)

    def quantize_pow2(v):
        return 2 ** F.round(F.log(v) / np.log(2.))

    n = get_parameter_or_create("n", (),
                                ConstantInitializer(n_init),
                                need_grad=True,
                                as_need_grad=not fix_parameters)
    xmax = get_parameter_or_create("xmax", (),
                                   ConstantInitializer(xmax_init),
                                   need_grad=True,
                                   as_need_grad=not fix_parameters)

    # ensure that bitwidth is in specified range and an integer
    n = F.round(clip_scalar(n, n_min, n_max))
    if sign:
        n = n - 1

    # ensure that dynamic range is in specified range
    xmax = clip_scalar(xmax, xmax_min, xmax_max)

    # compute step size from dynamic range and make sure that it is a pow2
    d = quantize_pow2(xmax / (2 ** n - 1))

    # compute min/max value that we can represent
    if sign:
        xmin = -xmax
    else:
        xmin = nn.Variable((1,), need_grad=False)
        xmin.d = 0.

    # broadcast variables to correct size
    d = broadcast_scalar(d, shape=x.shape)
    xmin = broadcast_scalar(xmin, shape=x.shape)
    xmax = broadcast_scalar(xmax, shape=x.shape)

    # apply fixed-point quantization
    return d * F.round(F.clip_by_value(x, xmin, xmax) / d)
```

### B.1.3 CASE U3: PARAMETRIZATION WITH RESPECT TO $d$ AND $q_{MAX}$

```python
def parametric_fixed_point_quantize_d_xmax(x, sign,
                                           d_init, d_min, d_max,
                                           xmax_init, xmax_min, xmax_max,
                                           fix_parameters=False):
    """Parametric version of `fixed_point_quantize` where the
    stepsize `d` and dynamic range `xmax` are learnable parameters.

    Returns:
        ~nnabla.Variable: N-D array.
    """
    def clip_scalar(v, min_value, max_value):
        return F.minimum_scalar(F.maximum_scalar(v, min_value), max_value)

    def broadcast_scalar(v, shape):
        return F.broadcast(F.reshape(v, (1,) * len(shape), inplace=False), shape=shape)

    def quantize_pow2(v):
        return 2 ** F.round(F.log(v) / np.log(2.))

    d = get_parameter_or_create("d", (),
                                ConstantInitializer(d_init),
                                need_grad=True,
                                as_need_grad=not fix_parameters)
    xmax = get_parameter_or_create("xmax", (),
                                   ConstantInitializer(xmax_init),
                                   need_grad=True,
                                   as_need_grad=not fix_parameters)

    # ensure that stepsize is in specified range and a power of two
    d = quantize_pow2(clip_scalar(d, d_min, d_max))

    # ensure that dynamic range is in specified range
    xmax = clip_scalar(xmax, xmax_min, xmax_max)

    # compute min/max value that we can represent
    if sign:
        xmin = -xmax
    else:
        xmin = nn.Variable((1,), need_grad=False)
        xmin.d = 0.

    # broadcast variables to correct size
    d = broadcast_scalar(d, shape=x.shape)
    xmin = broadcast_scalar(xmin, shape=x.shape)
    xmax = broadcast_scalar(xmax, shape=x.shape)

    # apply fixed-point quantization
    return d * F.round(F.clip_by_value(x, xmin, xmax) / d)
```

## B.2 Power-of-two quantization

### B.2.1 Case P1: Parametrization with respect to $b$ and $q_{\text{MAX}}$

```
1  def parametric_pow2_quantize_b_xmax(x, sign, with_zero,
2                                      n_init, n_min, n_max,
3                                      xmax_init, xmax_min, xmax_max,
4                                      fix_parameters=False):
5      """Parametric version of 'pow2_quantize' where the
6      bitwidth 'n' and range 'xmax' are learnable parameters.
7
8      Returns:
9          ~nnabla.Variable: N-D array.
10     """
11     def clip_scalar(v, min_value, max_value):
12         return F.minimum_scalar(F.maximum_scalar(v, min_value), max_value)
13
14     def broadcast_scalar(v, shape):
15         return F.broadcast(F.reshape(v, (1,) * len(shape), inplace=False), shape=shape)
16
17     def quantize_pow2(v):
18         return 2 ** F.round(F.log(F.abs(v)) / np.log(2.))
19
20     n = get_parameter_or_create("n", (),
21                                 ConstantInitializer(n_init),
22                                 need_grad=True,
23                                 as_need_grad=not fix_parameters)
24     xmax = get_parameter_or_create("xmax", (),
25                                    ConstantInitializer(xmax_init),
26                                    need_grad=True,
27                                    as_need_grad=not fix_parameters)
28
29     # ensure that bitwidth is in specified range and an integer
30     n = F.round(clip_scalar(n, n_min, n_max))
31     if sign:
32         n = n - 1
33     if with_zero:
34         n = n - 1
35
36     # ensure that dynamic range is in specified range and an integer
37     xmax = quantize_pow2(clip_scalar(xmax, xmax_min, xmax_max))
38
39     # compute min value that we can represent
40     xmin = (2 ** (-(2 ** n) + 1)) * xmax
41
42     # broadcast variables to correct size
43     xmin = broadcast_scalar(xmin, shape=x.shape)
44     xmax = broadcast_scalar(xmax, shape=x.shape)
45
46     # if unsigned, then quantize all negative values to zero
47     if not sign:
48         x = F.relu(x)
49
50     # compute absolute value/sign of input
51     ax = F.abs(x)
52     sx = F.sign(x)
53
54     if with_zero:
55         # prune smallest elements (in magnitude) to zero if they are smaller
56         # than 'x_min / \sqrt(2)'
57         x_threshold = xmin / np.sqrt(2)
58
59         idx1 = F.greater_equal(ax, x_threshold) * F.less(ax, xmin)
60         idx2 = F.greater_equal(ax, xmin) * F.less(ax, xmax)
61         idx3 = F.greater_equal(ax, xmax)
62     else:
63         idx1 = F.less(ax, xmin)
64         idx2 = F.greater_equal(ax, xmin) * F.less(ax, xmax)
65         idx3 = F.greater_equal(ax, xmax)
66
67     # do not backpropagate gradient through indices
68     idx1.need_grad = False
69     idx2.need_grad = False
70     idx3.need_grad = False
71
72     # do not backpropagate gradient through sign
73     sx.need_grad = False
74
75     # take care of values outside of dynamic range
76     return sx * (xmin * idx1 + quantize_pow2(ax) * idx2 + xmax * idx3)
```

### B.2.2   CASE P2: PARAMETRIZATION WITH RESPECT TO $b$ AND $q_{\text{MIN}}$

```python
def parametric_pow2_quantize_b_xmin(x, sign, with_zero,
                                    n_init, n_min, n_max,
                                    xmin_init, xmin_min, xmin_max,
                                    fix_parameters=False):
    """Parametric version of 'pow2_quantize' where the
    bitwidth 'n' and the smallest value 'xmin' are learnable parameters.

    Returns:
        ~nnabla.Variable: N-D array.
    """
    def clip_scalar(v, min_value, max_value):
        return F.minimum_scalar(F.maximum_scalar(v, min_value), max_value)

    def broadcast_scalar(v, shape):
        return F.broadcast(F.reshape(v, (1,) * len(shape), inplace=False), shape=shape)

    def quantize_pow2(v):
        return 2 ** F.round(F.log(F.abs(v)) / np.log(2.))

    n = get_parameter_or_create("n", (),
                                ConstantInitializer(n_init),
                                need_grad=True,
                                as_need_grad=not fix_parameters)
    xmin = get_parameter_or_create("xmin", (),
                                   ConstantInitializer(xmin_init),
                                   need_grad=True,
                                   as_need_grad=not fix_parameters)

    # ensure that bitwidth is in specified range and an integer
    n = F.round(clip_scalar(n, n_min, n_max))
    if sign:
        n = n - 1
    if with_zero:
        n = n - 1

    # ensure that minimum dynamic range is in specified range and a power-of-two
    xmin = quantize_pow2(clip_scalar(xmin, xmin_min, xmin_max))

    # compute min/max value that we can represent
    xmax = xmin * (2 ** ((2 ** n) - 1))

    # broadcast variables to correct size
    xmin = broadcast_scalar(xmin, shape=x.shape)
    xmax = broadcast_scalar(xmax, shape=x.shape)

    # if unsigned, then quantize all negative values to zero
    if not sign:
        x = F.relu(x)

    # compute absolute value/sign of input
    ax = F.abs(x)
    sx = F.sign(x)

    if with_zero:
        # prune smallest elements (in magnitude) to zero if they are smaller
        # than 'x_min / \sqrt(2)'
        x_threshold = xmin / np.sqrt(2)

        idx1 = F.greater_equal(ax, x_threshold) * F.less(ax, xmin)
        idx2 = F.greater_equal(ax, xmin) * F.less(ax, xmax)
        idx3 = F.greater_equal(ax, xmax)
    else:
        idx1 = F.less(ax, xmin)
        idx2 = F.greater_equal(ax, xmin) * F.less(ax, xmax)
        idx3 = F.greater_equal(ax, xmax)

    # do not backpropagate gradient through indices
    idx1.need_grad = False
    idx2.need_grad = False
    idx3.need_grad = False

    # do not backpropagate gradient through sign
    sx.need_grad = False

    # take care of values outside of dynamic range
    return sx * (xmin * idx1 + quantize_pow2(ax) * idx2 + xmax * idx3)
```

### B.2.3   CASE P3: PARAMETRIZATION WITH RESPECT TO $q_{\text{MIN}}$ AND $q_{\text{MAX}}$

```python
def parametric_pow2_quantize_xmin_xmax(x, sign, with_zero,
                                       xmin_init, xmin_min, xmin_max,
                                       xmax_init, xmax_min, xmax_max,
                                       fix_parameters=False):
    """Parametric version of `pow2_quantize` where the
    min value `xmin` and max value `xmax` are learnable parameters.

    Returns:
        ~nnabla.Variable: N-D array.
    """
    def clip_scalar(v, min_value, max_value):
        return F.minimum_scalar(F.maximum_scalar(v, min_value), max_value)

    def broadcast_scalar(v, shape):
        return F.broadcast(F.reshape(v, (1,) * len(shape), inplace=False), shape=shape)

    def quantize_pow2(v):
        return 2. ** F.round(F.log(F.abs(v)) / np.log(2.))

    xmin = get_parameter_or_create("xmin", (),
                                   ConstantInitializer(xmin_init),
                                   need_grad=True,
                                   as_need_grad=not fix_parameters)
    xmax = get_parameter_or_create("xmax", (),
                                   ConstantInitializer(xmax_init),
                                   need_grad=True,
                                   as_need_grad=not fix_parameters)

    # ensure that minimum dynamic range is in specified range and a power-of-two
    xmin = quantize_pow2(clip_scalar(xmin, xmin_min, xmin_max))

    # ensure that minimum dynamic range is in specified range and a power-of-two
    xmax = quantize_pow2(clip_scalar(xmax, xmax_min, xmax_max))

    # broadcast variables to correct size
    xmin = broadcast_scalar(xmin, shape=x.shape)
    xmax = broadcast_scalar(xmax, shape=x.shape)

    # if unsigned, then quantize all negative values to zero
    if not sign:
        x = F.relu(x)

    # compute absolute value/sign of input
    ax = F.abs(x)
    sx = F.sign(x)

    if with_zero:
        # prune smallest elements (in magnitude) to zero if they are smaller
        # than `x_min / \sqrt(2)`
        x_threshold = xmin / np.sqrt(2)

        idx1 = F.greater_equal(ax, x_threshold) * F.less(ax, xmin)
        idx2 = F.greater_equal(ax, xmin) * F.less(ax, xmax)
        idx3 = F.greater_equal(ax, xmax)
    else:
        idx1 = F.less(ax, xmin)
        idx2 = F.greater_equal(ax, xmin) * F.less(ax, xmax)
        idx3 = F.greater_equal(ax, xmax)

    # do not backpropagate gradient through indices
    idx1.need_grad = False
    idx2.need_grad = False
    idx3.need_grad = False

    # do not backpropagate gradient through sign
    sx.need_grad = False

    # take care of values outside of dynamic range
    return sx * (xmin * idx1 + quantize_pow2(ax) * idx2 + xmax * idx3)
```

