# OpenReview forum: "Mixed Precision DNNs: All you need is a good parametrization"
_ICLR.cc/2020/Conference — Accept (Poster)_

### Official Review · AnonReviewer1 · 2019-10-23
**Official Blind Review #1**

**Rating:** 6

**Review:**

This paper considers the problem of training mixed-precision models.
Since quantization involves non-differentiable operations, this paper discusses how to use the straight-through estimator to estimate the gradients,  and how different parameterizations of the quantized DNN affect the optimization process. The authors conclude that using the parameterization wrt the stepsize d and quantization range q_max has the best performance.

In the discussion for the three parameterization choices in section 2.1.
It is not clear how the range for U2 is obtained. Given d an integer, the gradient wrt b is also bounded. In this case, why is case U3 better than U2? In Table 1, it is shown that U2 also has good performance for uniform quantization.

Indeed, the gradient of any of the three parameters (stepsize, bitwidth and quantization range) can be derived by using chain rule given the gradients of the other two. It is not clear to me why some of them can be unbounded while others do not. In addition, It is not clear to me why having different gradient scales is a big problem. Adaptive learning rate methods like Adam should be able to help deal with the different scale of the gradients for three parameters. Can the authors compare the three parameterizations using Adam and see if similar empirical results can still be observed.

At the end of Section 2.1, the authors said that "similar considerations can be made for power-of-two quantization".  However, from table 1, these three parameterizations indeed have quite different performances for uniform and power-of-two quantization.  E.g., for uniform quantization, U2 and U3 perform significantly better than U1, while for power-of-two quantization, U1 and U3 perform significantly better than U2. Can the authors elaborate more on the difference?

Is the proposed differential quantization method used for both weight and activation? If so, how are the gradients w.r.t. the weights propagated through the quantized activations?

---------- post-rebuttal comment -------------
I thank the authors for their detailed response. It has solved most of my concerns and I accordingly raised my score.
---------------------------------------------------------

**Experience Assessment:**

I have published one or two papers in this area.

**Review Assessment: Checking Correctness Of Derivations And Theory:**

I assessed the sensibility of the derivations and theory.

**Review Assessment: Checking Correctness Of Experiments:**

I assessed the sensibility of the experiments.

**Review Assessment: Thoroughness In Paper Reading:**

I read the paper at least twice and used my best judgement in assessing the paper.

---

> ### Author Response · Authors · 2019-11-08
> **Response to Review#1**
>
> Thank you very much for your time and comments – please find below our point-to-point reply to each of them.
>
> "[...] in section 2.1. It is not clear how the range for U2 is obtained."
>
> To obtain the range of U2, we can have a look at eq. (3b). First, the derivative with respect to $q_{max}$ is bounded as the magnitude of the gradient is always smaller than one. For the derivative with respect to $b$, we know that $Q(x;\theta) – x$ can be bounded by $d/2$. Furthermore, the ratio $2^{b-1} / (2^{b-1} – 1)$ will be largest for $b = 2$ and, hence, the  derivative is in $[-d*log(2), d*log(2)]$ where $d$ for U2 depends on $b$ and $q_{max}$ via $d = q_{max} / (2^{b-1} – 1)$. We noticed that it is confusing that we use `$d$ in Eq. (3b) and will replace it by $q_{max} / (2^{b-1} – 1)$ in the final version of the paper.
>
> "Given d an integer, the gradient wrt b is also bounded. In this case, why is case U3 better than U2? In Table 1, it is shown that U2 also has good performance for uniform quantization."
>
> Please note that the step size $d$ does not need to be an integer but $d \in R^+$ (and it should be a pow2 for an efficient implementation). As we have shown above, the gradient can grow arbitrarily large as $d = d(b, q_{max})$ is not bounded.
> However, since a large $d$ is mostly not desired, you are right that exploding gradients are very unlikely in case of parametrization U2. U3 is superior to U2 mainly because the gradients with respect to $d$ and $q_{max}$ are decoupled for parametrization U3. This means that the derivative wrt $q_{max}$ is zero if the derivative wrt $d$ is non-zero and vice versa (as you can observe in eq. (3c)). Such a decoupling of the gradients is desirable for gradient-based optimization as we always optimize along conjugate directions, which is very effective.
>
> "Indeed, the gradient of any of the three parameters [...] can be derived by using chain rule given the gradients of the other two. It is not clear to me why some of them can be unbounded while others do not."
>
> As you you pointed out correctly, the derivatives of the three different parametrizations are related by the chain rule. The fact that some parametrizations have unbounded gradients is caused by the non-linear relationship of the parameters, i.e., $q_{max} = (2^{b-1}-1)d$. When converting the gradient of one parametrization to another, we will have to multiply with derivatives of this function. Note, that for example $\partial/\partial q_{max} b(d, q_{max}) = log(2) / ( q_{max} + d )$ might grow arbitrarily large for small $q_{max}$ and $d$.
>
> "Adaptive learning rate methods like Adam should be able to help deal with the different scale of the gradients for three parameters. Can the authors compare the three parameterizations using Adam and see if similar empirical results can still be observed."
>
> We also thought of this in our experiments, but did not include the results because of the page limit. In fact, our toy example in the appendix (e.g. page 12, Fig. 9) was run with the Adam optimizer. The performance difference is still considerable although an optimizer with adaptive learning rate was used. We will add this missing information to the appendix.
> The cause is mainly, that Adam needs the statistics of the gradients change smoothly over the parameter space to work well. If this is not the case, the estimated first and second moments of the gradients will be too noisy.
> Please note, that it also is not a simple scaling issue of the gradients, which can be solved by estimating the gradient magnitude and by normalizing it out. The gradient magnitude depends on the position in the quantization and weight parameter space, meaning that the gradient magnitude can explode for some parameter values (e.g. large $b$ for U1).
>
> "At the end of Section 2.1, the authors said that "similar considerations can be made for power-of-two quantization". [...] Can the authors elaborate more on the difference?"
>
> We did not intend to say that both uniform and pow2 quantizations have equal performance if we choose the right parametrizations. What we meant with “similar considerations” is, that for pow2 quantization, there are also three different parametrizations from which we can choose and that the parametrization that does not involve the bitwidth directly is better suited for optimization. Please note that the pow2 quantization scheme is much more restrictive as it constraints the weights to be pow2. This results in general in networks with worse performance compared to uniform quantization.
>
> "Is the proposed differential quantization method used for both weight and activation? If so, how are the gradients w.r.t. the weights propagated through the quantized activations?"
>
> Our method can be used to quantize both weights and activations. As we stated in Sec. 3 and 4, we used both activation and weight quantization in all of our experiments.
>
> We hope that we have answered all your concerns, we are welcoming any further discussion.

---

> > ### Author Response · Authors · 2019-11-08
> > **Results with ADAM optimizer**
> >
> > Reviewer#1 commented that solvers such as ADAM which scale the gradient for different parameters could eleviate the gradient scaling problem for parametrizations U1 and U2. Therefore, we have run additional experiments on CIFAR10 to collect more empirical evidence. We ran the training on CIFAR10 with ADAM optimizer (initial learning rate = 0.001) . Otherwise, we use the same experimental setup as we used for the SGD results presented in the paper.  We show the results for two cases: (1) Parameters quantization only and (2) Parameters and activation quantization.
> >
> > For Parameters Quantization only:
> >
> >        | SGD Momentum| ADAM    |
> > ==========================
> > U1  |    11.74%              |  7.61%  |
> > U2  |     7.44%               |  7.85%  |
> > U3  |     7.32%               |  7.36%  |
> >
> >
> > For Parameters and Activations Quantization
> >
> >       | SGD Momentum| ADAM    |
> > ===========================
> > U1 |      15.35%            |  7.54%  |
> > U2 |        7.74%            |  7.79%  |
> > U3 |        7.40%            |  7.40%  |
> >
> > These results are interesting because, to some extend, ADAM helps to cope with the 'bad' parametrizations (in particular for  the parametrization U1). Again, the parametrization U3 consistantly outperforms U1 and U2 for both weight and weight/activation quantization.
> >
> > We believe that this experiment further strengthens our main claim that parametrization U3 is superior to the other parametrizations. The main message is, that even if you use ADAM or momentum, the parametrization still matters a lot. We would like to thanks again the reviewer for this particularly interesting comment.

---

### Official Review · AnonReviewer3 · 2019-10-24
**Official Blind Review #3**

**Rating:** 6

**Review:**

The work studies differentiable quantization of deep neural networks with straight-through gradient (Bengio et. al., 2013). The authors find that a proper parametrization of the quantizer is critical to stable training and good quantization performance and demonstrated their findings to obtain mixed precision DNNs on two datasets,  i.e., CIFAR-10 and Imagenet.

The paper is clearly written and easy to follow. The idea proposed is fairly straight-forward. Although the argument the authors used to support the finding is not very rigorous, the finding itself is still worth noting.

One of the arguments that the authors used to support the specific form of parametrization is that it leads to diagonal Hessian. From optimization perspective, what matters is the condition number, i.e., max/min of the eigenvalues of the Hessian. Could this explain the small difference between the three different parametrization forms with uniform quantization and the big difference for power-of-two quantization?

The penalty method used to address the memory constraints will not necessarily lead to solutions that satisfy the constraints. The authors noted that the algorithm is not sensitive to the choice of the penalty parameters. Have the authors tried to tackle problems of hard memory constraints?



**Experience Assessment:**

I do not know much about this area.

**Review Assessment: Checking Correctness Of Derivations And Theory:**

I assessed the sensibility of the derivations and theory.

**Review Assessment: Checking Correctness Of Experiments:**

I assessed the sensibility of the experiments.

**Review Assessment: Thoroughness In Paper Reading:**

I read the paper at least twice and used my best judgement in assessing the paper.

---

> ### Author Response · Authors · 2019-11-09
> **answer to reviewer 3**
>
> Thank you for this interesting question about the differences between the uniform and the power-of-two quantization experiments.
>
> When comparing the gradients of the uniform quantizers in Eq. (3a-3c) to gradients of the power-of-two quantizers in Eq. (22-25), we can notice that the differences between the gradient scales is much larger for the power-of-two quantization. This means, that the scaling problem is more severe. You are correct, that this can lead to more ill-conditioned Hessians and, hence, could explain the bigger performance gaps between our three parametrizations in case of the power-of-two quantization. We think we can confirm this with some additional simulations.
>
> Your last question about the formulation of the constraint optimization problem is heavily related to the comments of reviewer 2. Therefore, it might be best to read our response to reviewer 2 at this point.

---

### Official Review · AnonReviewer2 · 2019-10-28
**Official Blind Review #2**

**Rating:** 6

**Review:**

The authors propose learning a quantizer for mixed precision DNNs. They do so using a suitable parameterization on the quantizer's step size and dynamic range using gradient descent, and where the quantizer's bitwidth are inferred from the former two rather than also learned jointly.

As a non-expert in the field, I found the paper well-written and interesting in their analysis of their proposed parameterization. They explain well how quantizers work, and the intuition and relationships of the parameters behind two popular types of quantizers: uniform and power-of-two. Equation (3) is especially explicit in understanding how the choice of 2 of the 3 parameters makes an impact on the choice of gradients. My understanding is that this is the core contribution.

Novelty-wise, I don't have enough background to tell if this is much of a leap from related work that has already proposed learning certain parameters of quantizers (but different parameters, or not the exact 2 proposed by the authors). I do like the discussion of related quantizer literature noted in the introduction.

I don't know if there is already previous work in the paper's follow-up section of learning quantized DNN under a constraint involving maximum total memory, total activation memory, and maximum activation memory. The solution of a Laplace multiplier seems fairly naive and hard to work in practice as it is not a hard constraint. As a naive question, how does the scale of these values compare to the original loss function? For example, if we think of the original loss function as a negative log-likelihood which computes bits/example, does it make sense to add a constraint penalty in kB as in the experiments, which is a completely different unit scale? Do you also backpropagate through the constraint function g?

**Experience Assessment:**

I do not know much about this area.

**Review Assessment: Checking Correctness Of Derivations And Theory:**

N/A

**Review Assessment: Checking Correctness Of Experiments:**

I assessed the sensibility of the experiments.

**Review Assessment: Thoroughness In Paper Reading:**

N/A

---

> ### Author Response · Authors · 2019-11-09
> **answer to reviewer 2**
>
> Thank you for your interesting comments concerning the formulation of the constraint
> optimization problem. We are delighted to see that you found our paper interesting and well written. We agree with you, that our approach to include the constraints in the loss function is straight-forward. However, this is not the main contribution of the paper and we would welcome any further work looking at better ways to do this.
>
> You are absolutely right, that optimizing our proposed cost function does not necessarily guarantee to actually achieve the desired constraint. Note that using a larger multiplier can help with this problem in practice, however the constraint term should not dominate the cost function.  As described in section 4, paragraph 2, we observed this issue when migrating from the CIFAR10 to the ImageNet experiments. As mentioned, it is important to choose the multipliers such that both the cost term (categorical cross-entropy in our case) and the penalty terms have comparable magnitudes after random initialization of the network. However, in practice, we observed that the performances are not sensitive to the choice of $\lambda_j$ as long as it is roughly scaled with the network size.
>
> For all experiments, we also back-propagate the error through g.

---

### Public Comment · ~Alessandro_Capotondi1 · 2019-10-21
**Mixed precision quantization through a rule-based approach**

Dear authors,

In our previous paper (https://arxiv.org/abs/1905.13082) we proved the effectiveness of another approach besides the ones you cited to select the quantization parameters.
In that work, we present a novel end-to-end methodology for enabling the deployment of low-error deep networks on microcontrollers. To fit the memory and computational limitations of resource-constrained edge-devices, we exploit mixed low-bitwidth compression, featuring 8, 4 or 2-bit uniform quantization, and we model the inference graph with integer-only operations. Our approach aims at determining the minimum bit precision of every activation and weight tensor are given the memory constraints of a device.
This is achieved through a rule-based iterative procedure, which cuts the number of bits of the most memory-demanding layers, aiming at meeting the memory constraints. After a quantization-aware retraining step, the
fake-quantized graph is converted into an inference integer-only model by inserting the Integer Channel-Normalization (ICN) layers, which introduce a negligible loss as demonstrated on INT4 MobilenetV1 models. We report the latency-accuracy evaluation of mixed-precision MobilenetV1 family networks on an STM32H7 microcontroller. Our experimental results demonstrate an end-to-end deployment of an integer-only Mobilenet network with Top1 accuracy of 68% on a device with only 2MB of FLASH memory and 512kB of RAM, improving by 8% the Top1 accuracy with respect to previously published 8-bit implementations for microcontrollers.

---

> ### Author Response · Authors · 2019-10-22
> **Mixed Precision: Rule based vs Learned bitwidth**
>
> Thank you for your interest in our work and for pointing at your paper, which also addresses the problem of finding the optimal bitwidth of per-tensor quantized DNNs. Your results for MobileNetV1 are interesting, as the networks reach good accuracy with a quite small memory footprint.
>
> While reading your paper and comparing it with our work, we see major methodological differences. You proposed an iterative offline algorithm to obtain the optimal bitwidth. To our understanding, the algorithm is based on a heuristic which reduces the bitwidth of those weight tensors that dominate the memory footprint of the network until it meets the memory constraint. Hence, the bitwidth selection can only depend on the network structure, but not on the data.  In contrast, our paper proposes a training method for quantized DNNs, where the bitwidth is learned in parallel to the network parameters. In other words our method optimizes jointly for accuracy and size. Therefore, our learned bitwidths is network structure as well as data dependent.

---

### Public Comment · ~Fabian_Timm1 · 2019-11-08
**Results require some clarification**

Thank you very much for your article. We are also working in the field of network quantisation and we really enjoyed reading.
The article is well structured and easy to follow. However, we kindly ask the authors clarify our major aspects and we really recommend to take care of our minor aspects.

-----------------------------------------------------------------------

Major aspects:
* Table 3: Where do you get the results of approach TQT for CIFAR-10? In the original TQT paper these values have not been mentioned. Did you recomputed them yourself? If so, which parameters did you choose?

* Table 3: Many other state-of-the-art approaches [1,2,3] also provide results for CIFAR-10, e.g. using VGG7 or DenseNet. So, why do you only compare with TQT?

* Table 4: Please, explain why the baseline error rate of 29.82% is larger than your best error rate of 29.41%. Same holds for ResNet-18, here your approach also improves the error rate of the pure floating point model (29.34% vs. 29.72%).

* Table 3+4: How do you obtain the baseline performance? Training from scratch or from other papers? If you did training from scratch, please provide the paramters for better comparison and reproducability.

* Table 4: The authors of the TQT paper evaluated their approach for 12 different networks on ImageNet. For comparison you used two of their performance values (MobileNet V2 and ResNet-18), why have you chosen those two networks?

* In equation 1 $\theta$ is defined by 3 values, in Fig. 1 $\theta$ takes only two values, why?

* Table 1 and 3: you provide results for ResNet-20 on CIFAR-10 in both tables, but where is difference? Should the error rates not be identical?

* Please, update your references. Especially over the past three years many papers in the field of quantisation have been published, e.g. [1,2,3]

* You intensively compare the error rates and the network size. Of course, the main focus is on quantisation and fast
inference on target hardware. But please also mention the training time in terms of epochs.

[1] Xiao Dong Chen, Xiaolin Hu, Hucheng Zhou, and Ningyi Xu. Fxpnet: Training a deep convolutionalneural network in fixed-point representation. IJCNN 2017.
[2] Fengfu Li and Bin Liu. Ternary weight networks.CoRR, abs/1605.04711, 2016.
[3] Lukas Enderich, Fabian Timm, Lars Rosenbaum, and Wolfram Burgard. Learning multimodalfixed-point weights using gradient descent. ESANN 2019.

-----------------------------------------------------------------------

Minor aspects (that would improve readability):
* Separate formulas and many mathematical definitions in the corresponding latex environment (of course this needs space). But using them within the text makes reading very hard, e.g. beginning of section 2.3

* Increase figures in size, e.g. legend of Fig. 3 is almost as large as the plots itself, same holds for Fig. 4

* Increase font size in Table 1, 2, and 3 - it is very hard to read

* In Sec. 2.3 2): you define $d_l$ with $W_l$, we guess $W_l$ is the number of weights in layer $l$; or is it the total number of weights?

* In Sec. 2.3 2): why do start with b=4bit? What happens if you use larger values for b?

We are really looking forward to your comments.

---

> ### Author Response · Authors · 2019-11-10
> **authors response  (part 1 of 2)**
>
> We are glad that our paper is of interest to you, and that you enjoyed reading it. Thank you for your very detailed and well structured list of comments.
>
> We noticed, that most of your comments concern the comparison of our approach to the related literature. As you said, many papers in the field of quantization have been published recently. Because of the length constraint, it is impossible to provide a full literature review of the DNN quantization literature in this paper. We therefore limited our discussion of the related work to: 1) Seminal papers such as Deep Compression (Han et al., 2015), 2) Papers which are thematically close to ours, i.e., which discuss how to learn the quantization parameters; eg. Wang et al., 2018 and  TQT (Jain et al., 2019), 3) Papers discussing other quantization methods that deliver state-of-the-art results for the same networks and tasks as we do. Thank you for pointing at your publication to ESANN 2019. We did not know about this paper and we will, of course, happily read it and add it as a reference if it is relevant to our paper.
>
> Concerning your list of questions:
>
> * The TQT method learns $q_{max}$, while the bitwidth $b$ is set manually. Therefore, we see it as a first step toward our approach which can learn all quantization parameters. The TQT results in Table 3 are obtained from our own implementation. We treated TQT as a special case, where we calculate the gradient with respect to $q_{max}$, but not with respect to $b$. This makes it directly comparable to our results.
>
> * We have chosen ResNet18 and MobileNetV2 because they are classical baseline methods in the quantization literature. We think that it makes most sense to compare the the effect of quantization, using networks which already have a small memory footprint (like ResNet18 or MobileNetV2). Compared to those models, the VGG type networks are heavily over parametrized and have a low efficiency, meaning that they obtain worse accuracies while having a larger computational complexity. We strongly believe that it would be very beneficial if the community working on network quantization would use the same set of baseline models as its foundation, performing the comparisons in the same way.  Unfortunatly this is not the case as of today each paper uses different tricks (e.g. quantizing only a subset of layers, etc.), what makes the comparison of the methods rather difficult. We would support any action toward creating a common benchmark for network quantization.
>
> * You are correct, that in our experiments on the ImageNet dataset, the best quantized networks reached a slightly better accuracy as the full precision baseline model. There are two reasons why this is possible: 1) We used the baseline model as an initialization when fine tuning with memory constraints, meaning that the quantized network actually has been trained considerably longer. 2) The memory constraints might act as a regularizer, which favors simple models with good generalization properties.
>
> * All baseline results are obtained, using our own implementation and trained from scratch. Our setup is as follows: SGD with momentum 0.9, learning rate 0.1; Fixed learning rate schedule with 3 drops by factor 0.1; input standardization; random flip with $p=0.5$; random crop.
>
> * You pointed out correctly, that in Eq. 1 we have 3 parameters, while in Fig. 1 $\theta$ only takes two values. Actually, this is related to the core idea of this paper. The reason why we can choose between three different parametrizations is, because 2 out of 3 parameters in $\theta$ are always dependent, meaning that we have ${3 \choose 2} = 3$ parametrizations with 2 independent parameters.

---

> > ### Author Response · Authors · 2019-11-10
> > **authors response (part 2 of 2)**
> >
> > * Table 1 gives the accuracy of the networks if we start from random initialization, using no hardware constraints. We have chosen this experiment, because only the parametrization of the quantizer has an influence on the results and not the formulation of the hardware constraint. It shows, that quantized DNNs can be trained from scratch, using STE gradients and that the parametrization U3 and P3 are optimal. In comparison, table 3 gives the accuracy when training with memory constraints, initializing the network weights from a pre-trained float32 network. We have chosen this experiment to show, that we can learn quantized DNNs which satisfy a given memory constraint, using the penalty method. Furthermore, it shows, that heterogenously quantized DNNs with a learned bitwidth have a performance superior to quantized DNNs with a homogenous and fixed bitwidth. So, both experiments use completely different setups and, hence, do not show the same results.
> >
> > * Of course, we can include the training time in the final version of our paper. Thank you for pointing it out.
> >
> > * We started from $b=4$ bit in all of our experiments, because in practice,  for $b>4$ bit it does not matter that much how a DNN is quantized. Even very simple offline algorithms (min/max quantization) will result in quantized DNNs with good performance.
> >
> > Of course, we will also try to resolve your minor aspects, addressing the formatting of the paper, when preparing our final version of the paper. However, due to the page limit, it might be hard to increase the size of all of the figures and tables.

---

### Author Response · Authors · 2019-11-15
**Manuscript update**

We would like to thank all reviewers and those who participated in the discussion for their thoughtful and helpful comments which helped us to improve our manuscript. In particular, we added the following revisions to the paper:

1) We added a comment why we start from $b=4$bit in Section 2.3.

2) We added a more detailed description of the experimental setup for the CIFAR-10 experiments in section 2.3 (including training time).

3) We added a paragraph to Section 3, discussing how to choose the penalty weights $\lambda_j$ when training networks with memory constraints.

4) We added a more detailed description of the experimental setup for the ImageNet experiments in Section 4.

5) We added that ADAM is used as an optimizer for the toy example in Section 2.3.

6) In Section 2, we added a discussion of the scaling issue. Especially, we discuss why this scaling issue is a problem even if we use adaptive gradient methods like ADAM.

7) We added a table to the appendix which compares the different quantization parametrizations when training a quantized ResNet-20 on CIFAR-10, using ADAM to give more empirical evidence. Please see Section A.4.

8) We replaced $d$ with $\frac{q_{max}}{2^{b-1}-1}$ in the discussion of the gradient norms in Section 2.1, for better readability.

---

### Author Response · Authors · 2020-05-20
**Release of NNabla source code**

Please find our NNabla source code here: https://github.com/sony/ai-research-code/tree/master/mixed-precision-dnns

---

### Decision · Program_Chairs · 2019-12-19

**Decision:**

Accept (Poster)

**Comment:**

The reviewers uniformly vote to accept this paper. Please take comments into account when revising for the camera ready. I was also very impressed by the authors' responsiveness to reviewer comments, putting in additional work after submission.